# Reconstruction of macroglia and adult neurogenesis evolution through cross-species single-cell transcriptomic analyses

David Morizet [1,2] ✉, Isabelle Foucher[1], Alessandro Alunni[1,3] & Laure Bally-Cuif [1] ✉

Macroglia fulfill essential functions in the adult vertebrate brain, producing and maintaining neurons and regulating neuronal communication. However, we still know little about their emergence and diversification. We used the zebrafish *D. rerio* as a distant vertebrate model with moderate glial diversity as anchor to reanalyze datasets covering over 600 million years of evolution. We identify core features of adult neurogenesis and innovations in the mammalian lineage with a potential link to the rarity of radial glia-like cells in adult humans. Our results also suggest that functions associated with astrocytes originated in a multifunctional cell type fulfilling both neural stem cell and astrocytic functions before these diverged. Finally, we identify conserved elements of macroglial cell identity and function and their time of emergence during evolution.

The appearance of nervous systems during animal evolution was a transformative event which revolutionized their interactions with the outside world and ultimately gave rise to higher cognition. To acquire their current forms, nervous systems required emergence of new cell types followed by substantial diversification[1]. To date, work on cell type evolution in the brain has focused on neurons. Conversely, macroglia have been mostly overlooked, despite being a major component of the human nervous system, outnumbering neurons in the cerebral cortex[2]. Macroglia fulfill essential roles to produce, guide and support neurons[3]. In mammals, the macroglia is made up of radial glia-like cells (referred to as radial glia -RG- below), astrocytes, oligodendrocytes and ependymocytes. RG are the only cells to act as neural stem cells (NSC) and support adult neurogenesis under physiological conditions, a process that promotes plasticity, growth and regenerative abilities in several species. The only glia in the nervous systems of early deuterostomes and early vertebrates were RG[4], but little is known about the actual heterogeneity of early glia and how glial diversity emerged. Additionally, the sequence of events of adult neurogenesis appears similar across vertebrates[5–7], yet a broad comparative analysis has never been conducted.

With its large RG population, abundant adult neurogenesis in regions homologous to mammalian neurogenic niches and intermediate glial diversity, the zebrafish telencephalon is an excellent model to address key questions of macroglia evolution and diversification[8]. Here we used scRNA-seq to profile all cell types of the adult zebrafish telencephalon and generate an improved characterization of its macroglia. We then re-analyzed data covering the telencephalon from several vertebrates[9–11] which included glia (even if glia had been discarded in the original analyses), or the nervous system of invertebrates[12–22], and used these to determine to which extent the populations of RG and the neurogenic cascade were conserved and how glial diversity emerged from an ancestral cell type. Overall, we generated single-cell transcriptomes for 17,710 cells and reanalyzed over 2,000,000 cells from other datasets in different species. These results improve our understanding of cell diversity in the zebrafish telencephalon, highlight conserved and divergent features of the neurogenic cascade across vertebrates, and reveal the multifunctional nature of ancestral astroglia.

## Results

### Generation of a molecular atlas of the adult zebrafish telencephalon

The zebrafish adult telencephalon is a major model for neuroscience studies, yet the limited characterization of its cell diversity hinders the labeling of specific cell types. Having previously characterized

[1]Institut Pasteur, Université Paris Cité, CNRS UMR3738, Zebrafish Neurogenetics Unit, Team supported by the Ligue Nationale Contre le cancer, F-75015 Paris, France. [2]Sorbonne Université, Collège doctoral, F-75005 Paris, France. [3]Present address: Institut des Neurosciences Paris-Saclay, Université Paris-Saclay, CNRS UMR9197, F-91190 Gif-sur-Yvette, France. ✉e-mail: david.morizet@pasteur.fr; laure.bally-cuif@pasteur.fr

the expression of several stem cell markers in the zebrafish pallium and found that *sox2* most comprehensively labels NSCs[23] we used fluorescence-assisted cell-sorting on cells dissociated from Tg(*sox2*:gfp) 3-month-post-fertilization (3mpf) adult fish (Fig. 1a) to enrich for RGs (located along the everted ventricular zone, green in Fig. 1a). We recovered and profiled both GFP+ and GFP- cells in equal numbers, to account for the possibility of *sox2*-negative NSCs and to better characterize cell diversity in the adult zebrafish telencephalon. We first grouped cells into broad classes using well-defined markers (Fig. 1b). We then subclustered each of those groups to identify refined cell subtypes. Because neurons and RG showed substantial heterogeneity we implemented a consensus-clustering strategy to resolve robust yet fine-grained clusters (Methods and Supplementary data Fig. S1a, b).

This analysis first revealed notable results on non-RG cell types (Fig. 1c). In accordance with previous work suggesting that homologs of the medial and lateral ganglionic eminences (MGE and LGE respectively) exist in the zebrafish telencephalon[24], we distinguished what likely corresponds to distinct ontogenies among GABAergic neurons, compatible with an origin from homologs of the MGE (*nkx2.1*+, *sox6a*+; inhi_1 to inhi_3), including somatostatin interneurons like in mammals, or the LGE (*six3a*+, *six3b*+, *meis2a*+; inhi_4 to inhi_8) (Fig. 1c). We also found that, based on their high expression of *nr2f1a* and *nr2f2*, a small proportion of GABAergic neurons likely derives from a homolog of the caudal ganglionic eminence (CGE; inhi_10 and inhi_11). Two recent studies aiming to characterize the teleostean telencephalon independently reached similar conclusions including regarding the potential existence of a previously undescribed CGE homolog with consistent molecular markers and position in the subpallium[25,26].

We also recovered unexpected diversity among brain immune cells despite a previous report suggesting that such heterogeneity was a human-specific trait[27]. Among *mpeg1.1*+ cells we identified two clusters corresponding to functionally and ontogenetically distinct subpopulations of microglia previously described in the midbrain[28]. We also identified another distinct and previously undescribed group of brain macrophages (Fig. 1c).

## Ventricular patterning is conserved throughout life and evolution

We next turned to quiescent RG (qRG) (Fig. 1b, green cluster). Conservative consensus-clustering followed by iterative cluster-merging based on differential expression identified 7 robust clusters (q1–q7) (Fig. 2a, Supplementary data Fig. S1c). We further confirmed that this was not dictated by technical parameters and that clusters could be re-identified with a classifier (Supplementary data Fig. S2), and identified cluster-specific gene signatures (Fig. 2b). There was little overlap between the clusters we identified and those proposed in a previous study[29]. We thus reanalyzed the previously published data and performed in situ hybridizations (ISH) on whole mount telencephala and on serial coronal slices to assess the validity of our results. Although we resolved fewer clusters in these reanalyzed data[29] than in our own, likely due to the significantly lower number of cells profiled in this previous study, the data structure and patterns of cluster-specific gene expression now appeared consistent between the two datasets (Supplementary data Fig. S3). ISH further corroborated our clustering as genes enriched in the same cells showed similar patterns (Supplementary data Fig. S4).

This analysis revealed spatially segregated RG populations in the ventricular zone. The *gsx2*+ q6 and the *nkx2.1*+ q7 clusters are restricted to a region close to the expected medial boundary between pallium and subpallium and in the ventral telencephalon respectively (Fig. 2c, d). Similar observations were made in adult murine lateral ventricles, where the lateral wall expresses *Gsx2* and is LGE-derived, while the ventral wall expresses *Nkx2.1* and is MGE-derived[10,30,31]. Markers for q5, including *nr2f1b*, were less strictly segregated but showed an enrichment caudally

in pallial RG (Supplementary data Fig. S5) reminiscent of the *Nr2f1* gradient in RG of the developing mammalian neocortex[32].

Thus, ventricular progenitor patterning is maintained not only throughout life but also throughout evolution. In particular, the ventricular zone of the dorsomedial pallium (Dm) in zebrafish, which has been the focus of most of the studies on telencephalic neurogenesis in this model, is homologous to the dorsal wall of the SVZ in mouse. Conversely the *gsx2*+ area corresponds to the subpallial Vd domain rather than Dm, contrary to what was previously reported[29], and to the lateral wall of the mouse SVZ. These molecular landmarks (Fig. 2e) will inform comparative studies of morphogenesis and neurogenic output between teleosts and mammals.

## Conserved and innovative features of adult neurogenesis

Adult neurogenesis has been mostly studied in rodents which display some of the highest genomic evolutionary rates and in which derived characters related to adult neurogenesis have already been previously identified[6]. Analyzing a broader range of species can improve generalizability and highlight fundamental features of the neurogenic cascade[5–7]. We recovered published datasets from amphibians[11,33], reptiles and mammals[10,34–39] and re-analyzed them using the same analysis pipeline as the one we had used for our own data, without going as far as systematic sub-clustering of broad cell classes. We used reported annotations and/or sets of co-expressed genes when available, which revealed that, despite independent filtering criteria, our analyses yielded results similar to the original publications. Next, we identified neuroblasts using markers such as *EOMES*, *NEUROD1* or *BHLHE22*, and RG using markers such as *GLUL*, *GFAP*, *FABP7*, and further subdivided RG into qRG and pre-activated RG (the latter corresponding to q1 in zebrafish) based on their transcriptional proximity to proliferating NSCs (Fig. 3a, b). We then compared gene expression patterns along these successive steps of neurogenesis progression to reconstruct their state in the last common ancestors of different taxa (Fig. 3c). This revealed conserved expression of key regulators since the last common ancestor of osteichtyes. For example, one Notch receptor gene is expressed in, and likely maintains, qRG[40,41]. Conversely, *NOTCH1* is expressed in pre-activated RG (paRG) and so is the Notch ligand gene *DLL1*[42,43]. The *SoxC* genes *SOX4* and *SOX11* are turned on in paRG and reach their peak in neuroblasts, consistent with their described roles in hippocampal neurogenesis[44]. Likewise, the EMT regulator *ZEB2*, which was recently found to be necessary for neurogenesis in the SVZ[45], displays conserved expression in neuroblasts. *SOX9*, which is a broad and specific marker of astroglia in adult mice[46] was not detected in previous datasets. We found that this is due to poor annotation of *sox9a* and that *SOX9* expression is conserved in qRG among Osteichthyes. We were unable to reconstruct neurogenic trajectories from large-brained mammals (pig and primates)[37,38], possibly owing to sensitivity limits of untargeted scRNA-seq, although we confirmed evidence of ongoing adult neurogenesis when reanalyzing a large macaque dataset[39]. While our re-analysis suggested that the initially defined RGL and IPC_2 clusters in this dataset are likely multiplets and of myeloid origin respectively (Supplementary data Fig. S6), these data did include progenitors not found in other datasets, allowing us to confirm the expression of some of the highly conserved genes in primates. In particular, *IGFBPL1* and *HES6* are highly and specifically expressed in IPCs and neuroblasts and thus represent promising markers to assess continued production of neurons in the human hippocampus (Supplementary data Fig. S6).

Conversely, we found variations of the neurogenic cascade over the course of evolution. ID transcription factors modulate the dynamics of Notch signaling. Among them *ID4* is more efficient than other IDs at inducing a return to quiescence[47], is the only one positively regulated by Notch signaling in qRG[48] and its expression in qRG is a tetrapod specificity (Fig. 3c). A switch between Notch receptors paralogs likely occurred early in the mammalian lineage, leading to *NOTCH2* being

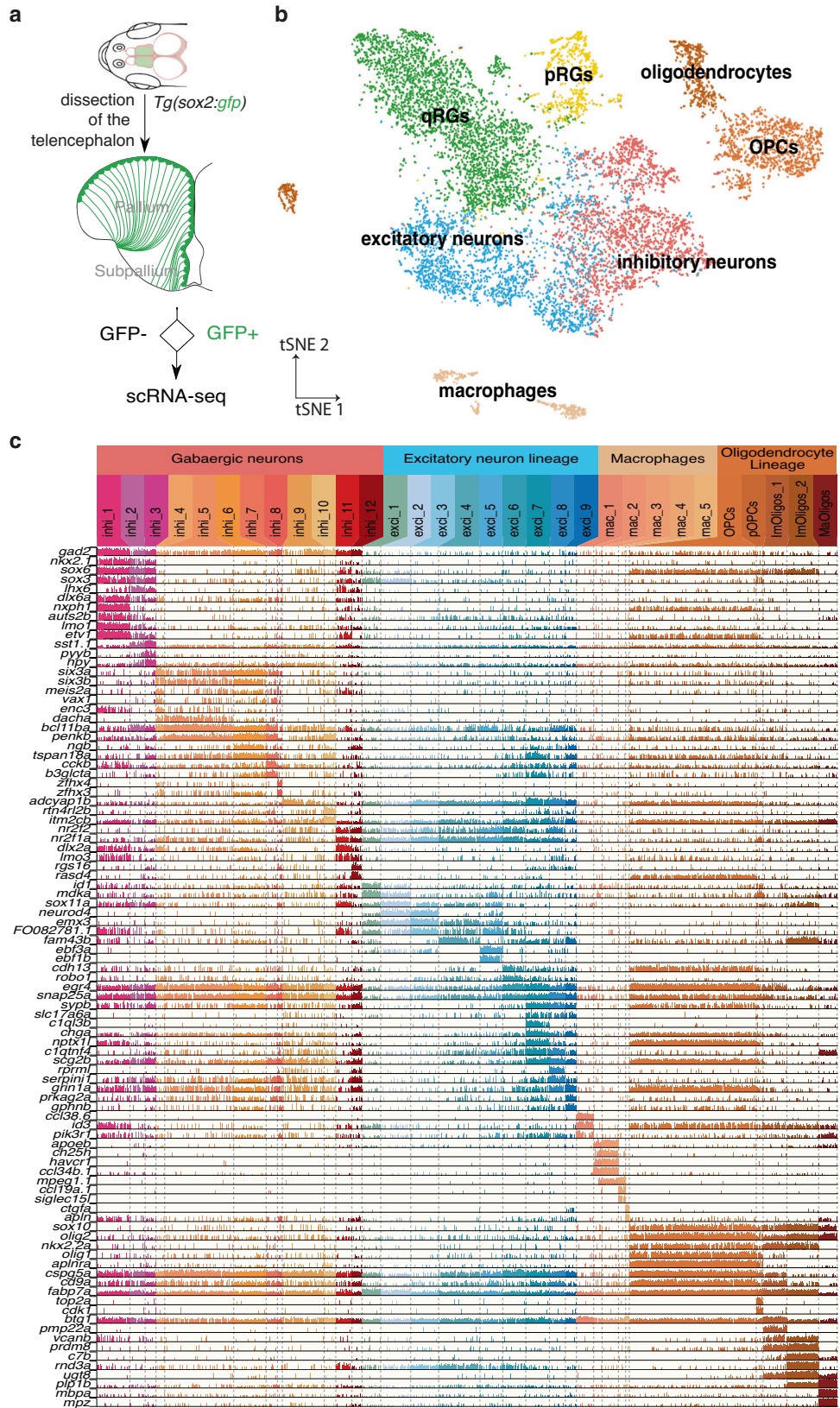

**Fig. 1 | The adult zebrafish telencephalon is characterized by an extensive cell type diversity. a** Cell collection strategy to enrich for RG. **b** tSNE of all cells colored by their broad cell type annotation. q quiescent, p proliferating. **c** Expression of marker genes in refined non-RG cell clusters. Each color-coded column is a cluster and each line is a gene. Each bar represents one cell and the height of the bar reflects the level of expression of the gene.

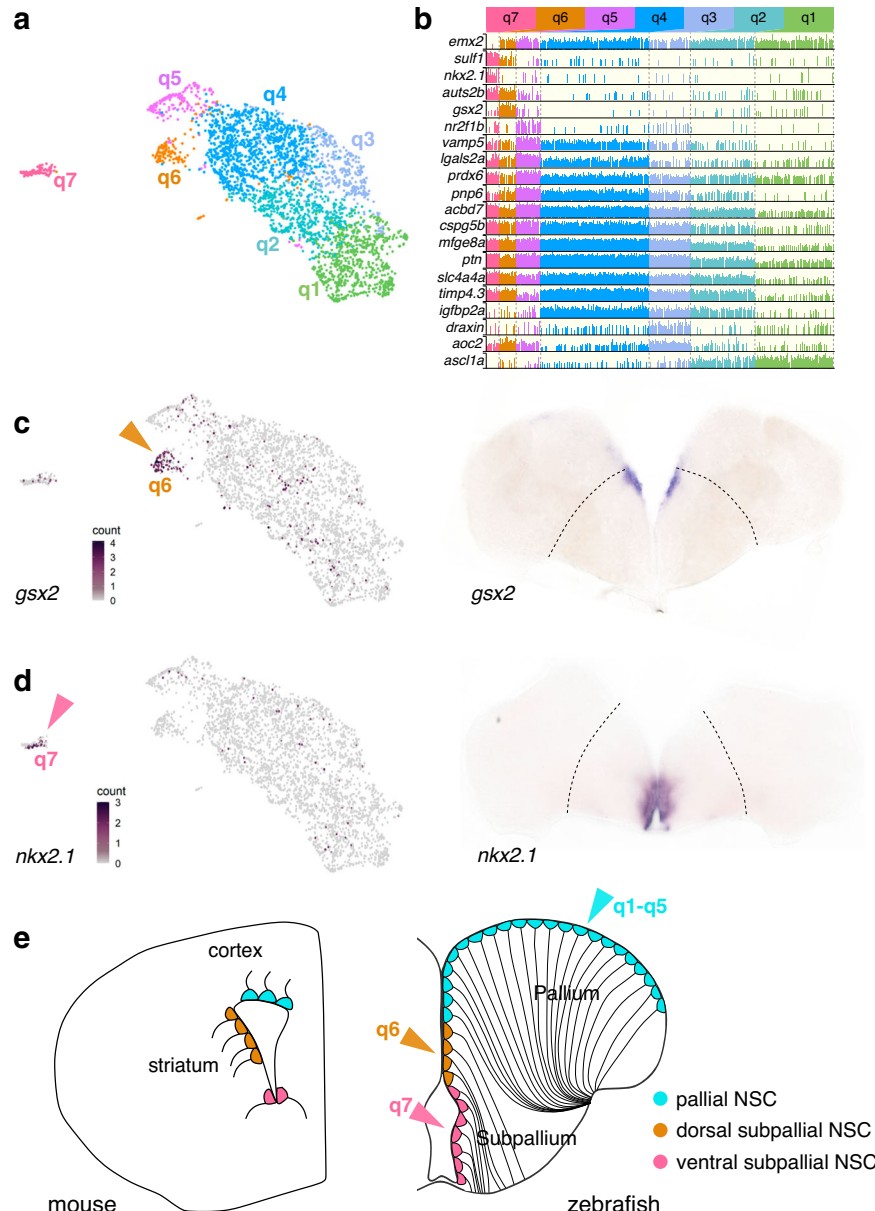

**Fig. 2 | qRG in the zebrafish adult telencephalon are spatially patterned.**
**a** UMAP of qRG (green cluster from Fig. 1b) colored by terminal cluster identity.
**b** Expression of marker genes across qRG clusters, highlighting cluster-specific, regional and quiescence markers. **c**, **d** Orthologs of ganglionic eminences markers (*gsx2*, *nkx2.1*) projected on zebrafish UMAP (left, color-coded arrows matching (a) and (b)) and with ISH on coronal slices of the adult zebrafish telencephalon (right).

Dashed lines indicate the boundary between pallium and subpallium. **e**, Homologies of ventricular territories between zebrafish and mouse telencephalon (coronal sections) inferred from expression of regionalized transcription factor genes such as *emx2*, *gsx2* and *nkx2.1*[10,94,95]. Depicted cells are RG, colored by their developmental origin (color-coded relative to clusters).

expressed instead of *NOTCH3* in qRG. In humans, *NOTCH2* and the human-specific *NOTCH2NL* genes, which potentiate Notch signaling and share the same promoter sequence, are co-expressed in RG during development[49,50], which could also account for some human-specific features of adult neurogenesis (Supplementary Text).

Together, this integrative approach highlights putative critical neurogenesis regulators either conserved throughout evolution or on the contrary responsible for taxon-specific properties.

### Evidence for quiescence depth heterogeneities in zebrafish qRG, and implications on the diversification of mammalian astroglia from ancestral RG

We found that our qRG clusters were further separated according to their quiescence depth (Supplementary data Fig. S7). Separation along

a quiescence to activation trajectory was also apparent in mouse SVZ datasets. We focused on a recent one which profiled a large number of cells from the different regions of the SVZ with high sensitivity[10]. To identify putative regulators of quiescence depth we developed a pseudo-ordering algorithm (Supplementary data Fig. S8A), applied it to both datasets and identified genes that show a strong association with deeper or shallower quiescence independently of region or species (Supplementary data Fig. S9). Some of these are likely part of a core quiescent stem cell gene set such as *HES1* which has been proposed to be a general marker of stemness[51], or those encoding the exosomal proteins CD9 and CD81 which are enriched in several types of quiescent cells[52].

Among the genes associated with deeper quiescence in zebrafish RG, several of them were expressed in both qRG and astrocytes in mice.

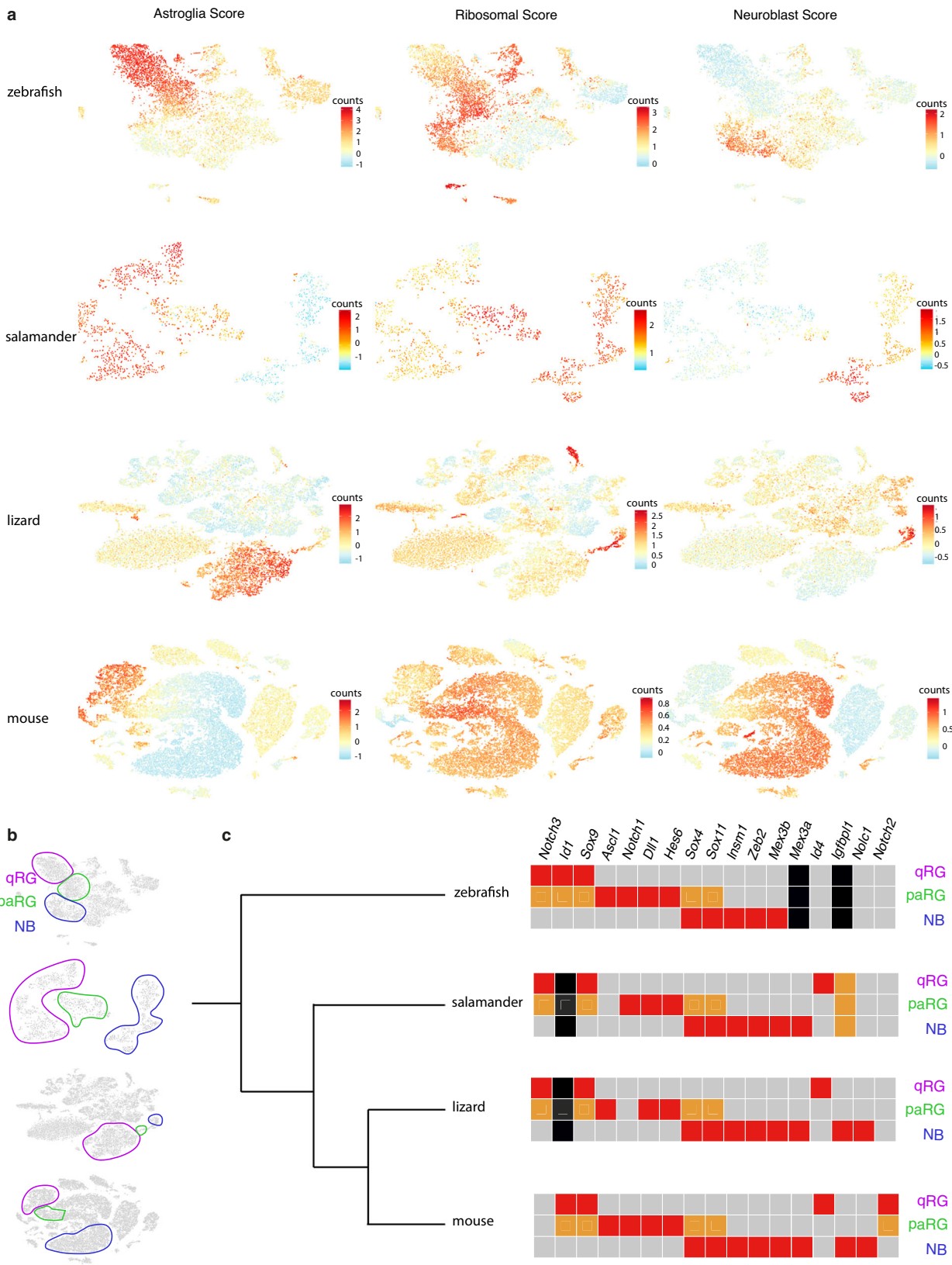

We also noticed that some of these genes were instead highly enriched in, or even specific of astrocytes across several datasets (Supplementary data Figs. S10–S13)[10,34,36,53]. This came as a surprise since besides mammals, most species are not thought to have astrocytes[54–56]. To further investigate the possible relationship between some qRG in zebrafish and mammalian astrocytes we focused on dorsal qRG (clusters q1 to q4,

Fig. 2a) to mitigate signals related to regionalization. We converted genes from zebrafish and mouse into a uniform nomenclature for orthologs (Methods) and mapped our pallial qRG to astroglia from mouse telencephalon using MetaNeighbor[57] (Fig. 4a). This revealed that similarity with astrocytes in zebrafish RG increases with estimated quiescence depth. Next, we scored mouse astroglial cells for genes

**Fig. 3 | Conservation and variations in the evolution of the adult neurogenic cascade in vertebrates. a** Plots displaying scores (see Materials and Methods) identifying astroglial cells, metabolically active cells and neuroblasts in zebrafish, salamander, lizard and mouse. The ribosomal score is used here to represent an upregulation in genes involved in protein synthesis, which is associated with a switch from quiescence to activation in stem cells, as well as with proliferation and early post-mitotic cells. Additional illustrations explaining how cell populations were selected can be found at https://entrepot.recherche.data.gouv.fr/privateurl. xhtml?token=bede1d62-f7cf-4e85-b6ce-05121176f108. **b** Schematic delineating which cell populations were selected for further analysis: in pink: qRG, identified by a high Glial Score and low Ribosomal and Neuroblast Scores; in green: paRG, identified by high Glial and Ribosomal Scores and a low Neuroblast Score, and in

blue: neuroblasts, identified by a low Glial Score and high Ribosomal and Neuroblast Scores. Proliferating cells were purposefully left out of this comparative analysis because non-glial progenitors and proliferating glia could not be readily identified in all datasets. **c** Phylogenetic tree depicting the profiled vertebrate species and reconstructed gene expression along the neurogenic cascade for selected genes. A grey square represents no or low expression, an orange square represents intermediate level of expression, and a red square represents high expression. Black squares represent genes that are either not present in the genome (*igfbpl1* in zebrafish) or for which it is impossible to distinguish between actual lack of expression or lack of detection due to scRNA-seq. NB: newly born post-mitotic neurons, paRG: pre-activated RG close to entering the cell cycle, qRG: quiescent RG.

---

enriched in q4 over q2. We chose these two clusters for our comparison because the staining patterns of cluster markers suggested that these two are intermingled, in particular in the most commonly studied Dm region, while q3 appears to be enriched more dorso-laterally and q1 may be biased by its proximity to proliferation and/or commitment. This revealed that astrocytes are enriched for q4 markers compared to RG (Fig. 4b, Supplementary data Figs. S10–S13). Gene ontology analysis of the genes that were enriched both in q4 over q2 and in astrocytes over RG confirmed that this list of genes encodes proteins involved in astrocytic support functions such as neurotransmitter synthesis and recapture, metabolic support, maintenance of ionic balance and modulation of ECM properties (Supplementary data Table S1). Several arguments further argue against the fact that astrocytic differentiation would simply be a maturation stage not attained in zebrafish radial glia, which would remain in some embryonic-like state. For example, zebrafish RG mature and acquire the q4 signature between the juvenile[58] and adult stage (Supplementary data Fig. S14a). Further, expression of the astrocytic geneset (Supplementary data Fig. S14B), and astrocytic differentiation and delamination, occur early during embryonic development of the mammalian brain rather than as an ultimate maturation step. Together, these results suggest that astrocytes evolved from ancestral RG through subfunctionalization in the mammalian lineage while astrocyte-like RG persist in zebrafish.

Next, we asked whether in zebrafish q4 astrocyte-like RG participate in adult neurogenesis or behave like radial astrocytes with no physiological NSC properties. The high level of transcriptome similarity among RG precluded lineage-tracing based on a specific q4 promoter. Instead, we analyzed the fate of clones from Dm genetically tagged with the Tg(*her4.3:ERT2CreERT2*) line[23] (Fig. 4c). Expression driven by this promoter fragment best matches *her4.1*, which is broadly expressed, enriched in q4 over q2 (Fig. 4d, compare *her4.1* with *timp4.3*), and cells from q4 could be sorted from Tg(*her4:egfp*) reporter fish using the same *her4* promoter. This suggests that, although clonal induction will likely not be specific to q4 cells, a substantial fraction of clones should originate from them. To verify this, we performed clonal induction and quantified *timp4.3* expression in recombined versus non-recombined cells using whole-mount RNAscope at 6 days post-induction (dpi) (the earliest time point to detect expression of the recombined reporter) (Fig. 4e, e' Supplementary data Fig. S15a,b2'). We found no statistical difference in *timp4.3* expression between these two cell groups, indicating that clonal induction distributed in a manner reflecting the proportion of q4 cells within the RG population (Fig. 4f, Supplementary data Fig. S15c, d). We reasoned that, after chase, the proportion of *her4*-driven clones that do not produce neurons, compared to the proportion of q4 cells, would thus allow us to infer whether q4 cells behave as NSC (Supplementary data Fig. S15e–g'). At 507 days of chase, we found that the proportion of RG-only clones was much lower than the proportion of q4 cells estimated in situ with RNAScope or in the scRNAseq dataset (less than 5% of RG-only clones vs consistent estimates centered around 43% for the proportion of q4 cells, p.value of two-sided binomial test < 1.5e10$^{-132}$) (Fig. 4g–i), confirming that these cells do have a constitutive neurogenic potential.

Together these results suggest that mammalian astrocytes emerged from ancestral neurogenic RG that already expressed many of the genes related to astrocytic-specific functions and that persist as a deeply quiescent but physiologically neurogenic RG population in zebrafish. Importantly, our results also suggest that stem cells can perform differentiated cell functions when they are quiescent, contrary to the common view that quiescent stem cells are inactive.

### Evolution of astrocyte-like cells across Planulozoa

Cell types similar to astrocytes appeared multiple times throughout evolution. Among vertebrates, the only class other than mammals where astrocytes are the major astroglial cell population are birds. Absence of astrocytes appears to be the ancestral state in reptiles, and turtles —the closest living relative to birds and crocodilians— do not have astrocytes either, suggesting that the cells described as astrocytes in birds evolved independently from those in mammals. We thus asked whether avian and mammalian astrocytes displayed a similar signature. We found that datasets generated from the high vocal center of zebra finch included both astrocytes and RG[59], with a substantial overlap between the top genes separating astrocytes from RG in birds and mammals (Supplementary data Fig. S16), suggesting that avian and mammalian astrocytes likely evolved in a similar way, from the same ancestral RG population.

The expression of a conserved astrocytic gene set in some zebrafish RG suggests that it emerged before the individualization of astrocytes as a cell type. Moreover, parenchymal glia associated with support functions are present in all major branches of bilaterians[60]. To estimate the time of emergence of this astrocytic gene set we asked whether its genes were expressed across Planulozoa by recovering and analyzing datasets from over 20 species[9–22,38,39,53,59,61–63], identifying existing orthologs and assessing their expression in glial clusters or among ectodermal cells (Fig. 5). We found expression of the astrocytic gene set in all vertebrate RG, but not in Ciona ependymoglia (Supplementary data Fig. S17). Ascidians may have undergone secondary nervous sytem simplification, thus we also attempted to analyze the cephalochordate Amphioxus to distinguish between acquisition of an astrocytic gene-set in vertebrates or in early chrodates with secondary loss in Ascidians. However, the available adult brain datasets to this day[64,65] were designed to collect many cells and identify broad cell classes and thus do not provide the necessary resolution to draw reliable conclusions. On the other hand, we did not find any group of cells co-expressing genes from the astrocytic gene set in cnidarians or protostomes except in insects, where distinct glial subpopulations expressed part of the astrocytic gene set. Ensheathing glia, and to a lesser-extent astrocyte-like and perineurial glia, co-expressed several genes associated with mammalian astrocytes (Supplementary data Fig. S18). In addition, glial cells in protostomes did not express homologs of *SOX2* or *SOX9* genes, which are part of a conserved astroglial gene regulatory network in vertebrates.

This large-scale comparative analysis suggests that while avian astrocytes are homologous cells to mammalian astrocytes, glial cells described in protostomes are not. Several genes involved in astrocytic

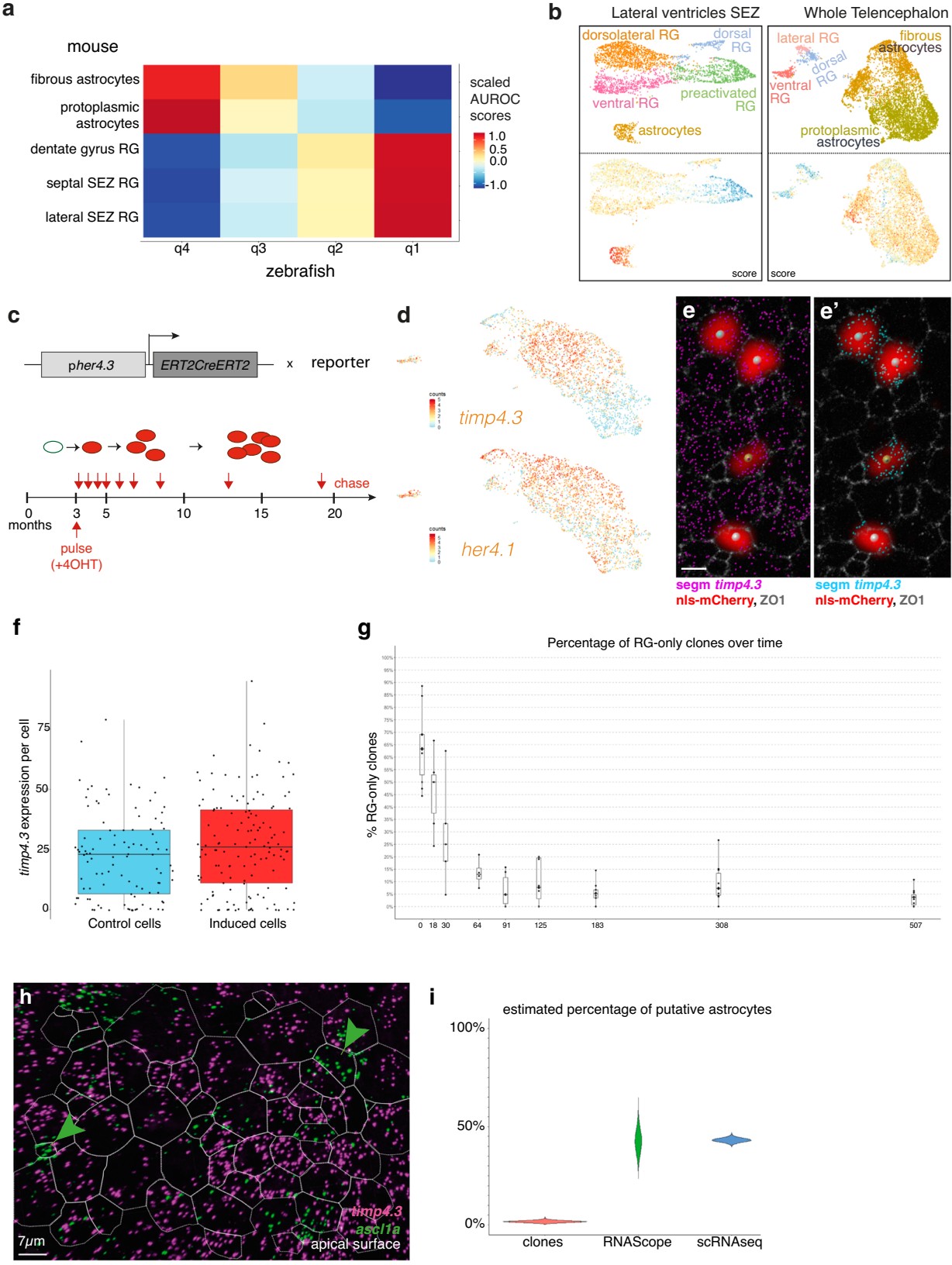

functions started being expressed in vertebrate RG and make up an astrocytic synapomere[66]. The emergence of parenchymal glia in insects, which together express many of the genes of the astrocytic synapomere, is likely the result of homoplasy, suggests remarkable functional convergence and highlights the interest of studying aspects of glial physiology in protostomes.

## Evolution of other macroglial cells

Cells of the oligodendrocytic lineage and ependymocytes make up the macroglia alongside astroglia. We used our zebrafish telencephalon dataset to identify cells belonging to the oligodendrocytic lineage and compared expression of putative core regulators and effector genes across chordates (Supplementary data Fig. S19). Others recently

**Fig. 4 | Transcriptomic and functional homologies between quiescent NSCs in zebrafish and mammalian astrocytes. a** Cluster mapping between zebrafish dorsal RG (this study) and astroglia in the mouse telencephalon. Colors: scaled AUROC scores. **b** Scoring of mouse astroglial cells from[10] (left) and[53] (right) for genes enriched in zebrafish q4 over q2. Top: annotated clustering of astroglia from each dataset. Bottom: Enrichment score for orthologs of genes overexpressed in zebrafish q4 over q2. **c** Scheme of the clonal recombination and chase times used to assess the neurogenic potential of q4 RG. **d** UMAP comparing *timp4.3* and *her4.1* expression in qRG. Colors: variable counts. **e, f,** Quantitative analysis of *timp4.3* expression in clones at 6dpi. e,e', Representative image of recombined cells (mCherry immunohistochemistry -IHC-, red, large spot) in Dm (ZO1 IHC, white, to delimit apical NSC surfaces), with whole-mount RNAscope for *timp4.3*. *timp4.3* dots are segmented in all apical cells (**e**, magenta) or shown only in recombined clones (e', cyan). Scale bar 5 µm. **f** *timp4.3* expression (number of dots) in non-recombined

(control) vs recombined (induced) cells at 6dpi (see Source Data file). For each boxplot: lower and upper bounds: 25th and 75th percentiles, internal line: median, whiskers extend to the extrema. Individual values overlaid as black dots. *n* = 135 and 103 for induced and control cells respectively. Statistical comparison: Two-sided Mann-Whitney U test, p:0.07; effect size: Cliff's delta (0.14) with a 95% confidence interval (−0.01, 0.28). **g** Proportion of clones per hemisphere containing only RG over time. Boxplots drawn as in f. *n* = 10, 6, 5, 4, 6, 11, 9, 10, 11 clone-containing hemispheres at 6, 18, 30, 64, 91, 125, 183, 308, 507 days respectively (see Source Data file). **h** Representative image of smISH for *ascl1a* (green) and *timp4.3* (magenta) in adult zebrafish Dm (dorsal view). ZO1 IHC delimits apical NSC surfaces (dotted lines). Scale bar 7 µm. **i** Bootstrapped estimate of the proportion of RG-only clones after a 507-day chase (red), versus proportions of q4 cells, estimated in situ with RNAScope (green) or in scRNAseq (blue).

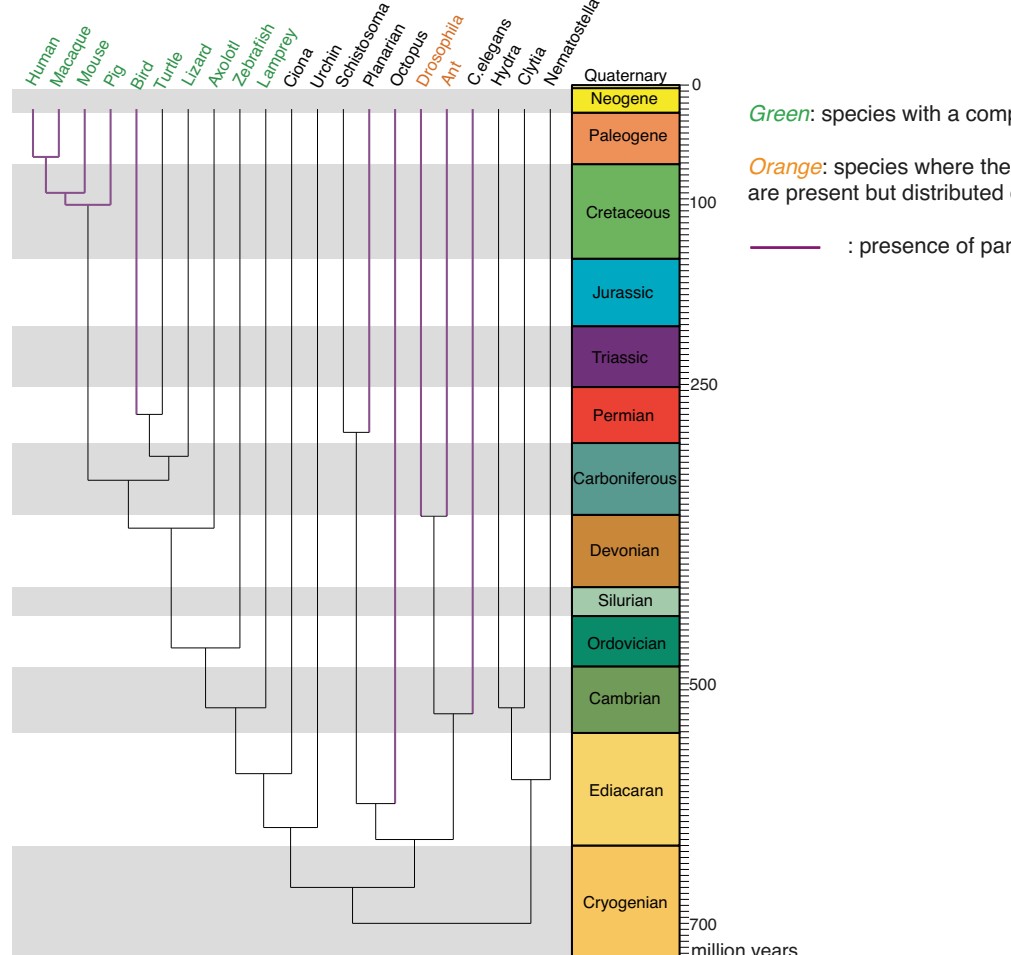

**Fig. 5 | Emergence of the astrocytic synapomere.** Phylogenetic tree depicting the expression of the astrocytic synapomere in analyzed species and whether parenchymal glia with supportive functions have been described in those species. Leaves in magenta represent phyla in which parenchymal glial cells have been previously described. Species in green co-express several genes of the astrocytic

synapomere in the same glial cell clusters. Species in orange express several genes from the astrocytic synapomere but spread out across several glial cell clusters. Species in black do not seem to rely on the astrocytic synapomere. Time scale in million years.

proposed that cells expressing *sox9b* are zebrafish astrocytes[67]. We found that they are transcriptionally closer to cells of the oligodendrocytic lineage and express several genes associated with oligodendrocyte maturation such as *c7b, rnd3a, ugt8, cadm4, gpr17, myrf* and *plp1b*, as well as low levels of *mbpb*. These cells might thus correspond to maturing oligodendrocytes, similarly to a population in the larval spinal cord[68]. For inter-species comparisons, we grouped cells based on the expression of *CSPG4* (and/or *PDGFRA* for tetrapods). A core set of transcription factors made up of *NKX2.2, SOX10, OLIG2* and *OLIG1* is

largely conserved across vertebrates. *NKX2.2* and *SOX10* are also expressed in lamprey glia, although orthologs for *olig* genes have not been identified in the lamprey genome. *CSPG4* (or *NG2*) which is commonly used to label oligodendrocyte progenitors (OPCs) is also widely conserved across vertebrates and detected in lamprey glia, contrary to *PDGFRA* and *APLNRA* which are restricted to tetrapod and zebrafish OPC respectively. The expression of genes involved in myelin formation appears variable, with the exception of *MBP* and *PLP1*. In scRNA-seq studies, only low levels of *plp1a* were detected whereas a

recent proteomic study found that PLP1B is 86 times more abundant[69]. We found that this discrepancy results from an incomplete gene model for *plp1b* which is indeed expressed at high levels in zebrafish oligodendrocytes. Lamprey glia express not only *PLP1* but also *MPZ*, like jawed fish, suggesting that co-expression is the ancestral state, lost in tetrapods. Despite being commonly associated exclusively with PNS, myelin *PMP22* was detected in most species besides mouse. *PMP22* being restricted to the PNS is thus a derived trait in mice. The core gene set conserved across vertebrates is not present in *Ciona* suggesting that the emergence of glia with oligodendrocytic properties happened in vertebrates after they split from other chordates.

Ependymocytes likely appeared as a cell-type in the telencephalon later in vertebrate evolution. In the zebrafish telencephalon similar cells are restricted to a rostral and ventral area[70] and were not profiled in our dataset. We detected a few in lizard and birds[9,59] where they have been observed in small numbers along the ventricles whereas in mammals they cover the whole ventricular surface and represent a much larger fraction of astroependymoglial cells in the niche. Ependymocytes express several markers of qRG, but simultaneously display remarkable convergence with other multiciliated cells which appear to have co-opted a similar multiciliated cell apomere (Supplementary data Fig. S20a). We found that mouse ependymocytes express regulators of SVZ neurogenesis that are not detected in lizards or birds, suggesting that they acquired additional regulatory roles in mammals (Supplementary data Fig. S20b).

## Discussion

Our molecular and cellular characterization of the adult zebrafish telencephalon expands knowledge on telencephalic evolution and will facilitate targeted studies of its diverse cell populations. In particular, we identify underappreciated heterogeneity among brain macrophages and conserved ontogenies of GABAergic neuron families across vertebrates.

In addition, using a custom consensus-clustering approach we identified distinct robust subpopulations of qRG. This is independent of the specific genome assembly, feature choice or clustering parameters and is reproducible across datasets and supported by subsequent cross-species comparisons and in situ validation. Heterogeneity among RG is explained to some extent by spatial origin, with homologous territories between mouse and zebrafish maintaining their positional identity from development to adulthood.

The presence of a caudo-rostral gradient of *nr2f1b* shows that this gradient predates neocortex arealization. The organization of the zebrafish pallium is less well-defined than that of the neocortex and appears to have evolved independently[71], suggesting that *NR2F1* might have been co-opted to specify sensory areas[32] after the split between actinopterygians and sarcopterygians. Of note, expression of *wnt3a* has been shown to be restricted to the lateral pallium in zebrafish[72]. Therefore, both caudo-rostral and medio-lateral gradients involved in hippocampus specification were already present in the last common bony fish ancestor. In agreement with this, transcriptomic studies suggest that the teleost dorsolateral pallium harbors neurons that share molecular signatures with murine hippocampal neurons and other tetrapod homologous regions[25,26].

Comparisons of adult neurogenesis across vertebrates revealed a highly conserved neurogenic cascade. Besides generating a list of putative critical regulators of adult neurogenesis in vertebrates, our approach highlights core principles of neurogenesis shared across phyla. Stepwise additions to this conserved core occurred during evolution and likely underlie specific properties of adult neurogenesis and NSC maintenance in different species, which can now be tested via anachronistic gene expression manipulation. This is not to say, however, that because the molecular pathways involved in neurogenesis are highly conserved throughout evolution, neurogenesis itself is necessarily a universally conserved feature. In particular, a switch from

*NOTCH3* to *NOTCH2* in mammalian qRG, together with the emergence of functional *NOTCH2NL* genes in humans, provides a realistic and testable hypothesis for the premature depletion of RG in humans (Supplementary Text). We also identify genes that show a conserved association with quiescence depth, including some showing consistent patterns in other tissues and/or cancers. Further investigation of those candidates is thus likely to yield fundamental insights on the maintenance of stemness and quiescence.

Our data suggest that avian and mammalian astrocytes evolved from the same ancestral multifunctional RG and that similar RG persist in zebrafish. The synapomere associated with those sister cell types is enriched in genes involved in metabolism, neurotransmission fine-tuning and extracellular milieu homeostasis. Several of these genes can also be detected in lamprey glia but not in putative *Ciona* ependymoglia. Likewise, the genes that segregate in oligodendrocytes in gnathostomes and are expressed in lamprey glia can induce intricate membrane re-organization and prevent axonal degeneration. Together, these observations are consistent with an ancestral role in enabling neuron communication and survival via the establishment of structures that later served as templates for the development of other functions such as myelination with which we usually associate macroglia. Macroglia then diversified through subfunctionalization, division of labor, co-option of new modules and neofunctionalization.

We expected to detect cells that retained expression of a similar gene set across bilaterians but were unable to do so in non-deuterostomes besides insects. Although we attempted to maximize sensitivity by surveying many datasets and including only those with high coverage, this approach can produce false negatives for rare cell populations or due to imperfect genomic annotations. Alternatively, it is possible that the functions associated with the astrocytic synapomere were initially performed by other cells including neurons themselves and became the property of glial cells in vertebrates. In that case, even if a common set of ancestral glia existed before they might not have expressed genes enriched in current vertebrate macroglia and would have been missed with our approach. Additional targeted studies will be helpful to shed light on the origins and evolution of glia, their coevolution with neurons and will benefit from focusing on genes identified in this study.

Finally, the presence of astrocytic-like RG in vertebrates that do not have bona fide astrocytes, such as the zebrafish, has several implications, including on the notions of quiescence and dedifferentiation (Supplementary Text). Moreover, it raises interesting questions regarding the evolution and other potential functions of these cells. Are they also neurogenic in salamanders and reptiles? How do they behave upon injury and to what extent do they contribute to regeneration? Understanding the behavior of these cells, which look like intermediates between RG and astrocytes, could facilitate reinstatement of stem cell properties in astrocytes to promote regeneration.

## Methods

### Fish lines and maintenance

All procedures relating to zebrafish (*Danio rerio*) care and treatment conformed to the directive 2010/63/EU of the European Parliament and of the council of the European Union. The animal study protocol was approved by the Ethics Committee n°39 of Institut Pasteur (authorization #36936, April 26th, 2022) and DDPP-2021-021 of the Direction Départementale de la Protection des Populations de Paris. Zebrafish were kept in 3.5-liter tanks at a maximal density of five per litter, in 28.5 °C and pH 7.4 water. They were maintained on a 14-h light/10-h dark cycle (light was on from 8 a.m. to 10 p.m.) and fed three times a day with rotifers until 14 days post-fertilization and with standard commercial dry food (GEMMA Micro from Skretting) afterwards. All fish used for experiments were between 3 and 4 months old, and sexes were mixed (information on this parameter was not collected as no

differences in adult pallial RG were noted between sexes). Fish from the transgenic line *Tg(sox2:GFP)*[73] and all other fish used for stainings were maintained on an AB background.

## Euthanasia

Fish were euthanized in ice-cold water (temperature comprised between 1° and 2 °C) for 10 min, according to a special dispensation and following the guidelines of the Ministry of Superior Education, Research, and Innovation.

## Cell dissociation and cell sorting

Cell sorting was conducted three days in a row to collect replicates, using twenty 3 months old adults from the *Tg(sox2::GFP) line*[73] on each day. Brains were dissected in Ringer's solution. The telencephalon was separated from the midbrain and the olfactory bulbs were removed. The two hemispheres were separated and cut along the boundary between pallium and subpallium to enrich for pallial cells. Cell dissociation was carried out according to[74] Briefly, brains were dissected in petri dishes filled with Ringer solution and placed on ice. They were cut into small pieces and collected in 1.5 mL tubes containing 1 mL 4 °C Ringer solution. The Ringer solution was removed and replaced with 500 μL FACSMax cell dissociation solution and incubated at 30 °C for 5 min. Afterwards dissociation was completed by pushing the resulting solution through a 40 μm cell-strainer and recovered in 10 mL ice cold PBS 1X solution in a new 15 mL tube. Dissociated cells in PBS were centrifuged at 400 g for 5 min at 4 °C and the supernatant was replaced with an appropriate volume of PBS to proceed with FACS. Cells were then sorted on a FACSAria III. We used forward and side scatter and DAPI staining to distinguish live cells from debris and sorted cells on their GFP levels to enrich for *sox2*-expressing cells while still including GFP negative cells to not miss any relevant population. After sorting, encapsulation of cells and reverse transcription was immediately performed using the 10x Chromium Controller and Chromium Single Cell 3' Kit v2 and then immediately frozen at -80 °C until all replicates had been collected.

## Library preparation and sequencing

After reverse transcription all replicates were processed in parallel with the v2 Kit. After barcoding libraries were pooled and split over multiple lines of a HiSeqX and sequenced at a depth over 100k reads per cell using 2 × 150 paired-end kits with the following sequencing read recommendation Number of Cycles: 26 cycles Read1 for cell barcode and UMI, 8 cycles I7 index for sample index and 98 cycles Read 2 for the transcript. This yielded a saturation above 95% for all libraries.

## Mapping and filtering of data

Initial analysis was conducted using the cellranger software. Reads were demultiplexed and mapped to the GRCz11 zebrafish genome assembly from Ensembl, although we later replicated our analyses with a genome assembly curated by the Lawson lab[75]. Datasets were first analyzed individually to determine whether they were fit for integration. First, we filtered cells based on the number of genes (nGene) and UMIs (nUMI) detected as well as on the relationship between nGene and nUMI. nGene is expected to be positively correlated with nUMI. Lower nGene than expected for a given nUMI value suggests low library complexity whereas higher nGene than expected for a given nUMI value suggests excessive library complexity and likely doublets. To filter on that criterion, we fit a loess regression curve for nGene-nUMI with a span of 0.5 and of gaussian family and removed cells which residuals were beyond three standards deviation of the mean[76]. Then we inspected the frequency of cells associated with a given nGene. This yielded a distribution with a narrow peak at low levels of nGene followed by a broader distribution centered around a peak close to nGene = 900. We considered the first peak to be low quality cells and set a threshold to remove it from the rest, which ended up

being nGene = 200. We also removed cells which had abnormally high nGene or nUMI based on the distribution on a plot of nGene as a function of nUMI. Then we removed genes expressed in fewer than 10 cells. Finally, we computed the percentage of the transcriptome that consisted of mitochondrial gene (percent.mito) and inspected plots of percent.mito as a function of nGene. These two variables show negative correlation because high percent.mito tend to be observed in low quality cells, in which fewer genes are detected. We set a threshold at 10% which removed a tail of cells with high percent.mito and low nGene. Subsequent inspection of the variation of gene expression as a function of technical variables such as nGene after regular scaling and normalization revealed that variance in gene expression was not dependent on technical factors. This pre-integration analysis also suggested that the second replicate was of lower quality than the other two which was consistent with the fact that it took longer to go from cell sorting to cell capture. Thus we only used replicates 1 and 3 for subclustering analyses to avoid clusters driven by technical artifacts.

## Identification of clusters and their markers

We did not need to use any batch removal method to obtain good merging between the replicates. Variable genes were identified by selecting genes exhibiting high variability given their level of expression without using a fixed general threshold. We selected PCA components to include based on whether they explained over 1% of the variance across the first 100 principal components and whether that variance was flagged as significant by the JackStraw test[77]. Initial clustering was performed on all the cells using a simple Smart Local Moving algorithm[78]. We looked for and removed doublet clusters using three separate approaches. The doublet cells methodology from the scran package[79] was used to score individual cells and the doublet cluster methodology to detect clusters which look like a mixture of two other clusters. An adapted DoubletFinder algorithm was used to generate doublets from randomly selected pairs of cells and detect cells which frequently co-clustered with those mock doublets[80]. This multi-pronged approach ensured that we did not keep any artefactual cluster at this stage.

Several broad cell types were already subdivided into multiple clusters. We isolated each individual lineage and performed further clustering to identify fine-grained cell subpopulations. Substantial heterogeneity was apparent in neurons and RG. To ensure that we did not overcluster we developed a consensus clustering approach. We pooled results from Bayesian model mixture, smart local moving algorithm, multilevel algorithm, walktrap, spinglass and density peaks. kNN graphs used for the graph clustering approaches were themselves built using edges weighed either as in SNN-Cliq[81] or Phenograph[82]. From this we obtained a consensus matrix with the Cluster-based Similarity Partitioning Algorithm[83]. From this consensus matrix we drew a final hierarchical clustering and used a conservative cutoff to identify robust clusters. We then subjected each pair of neighboring clusters to differential gene expression detection to determine whether they should be merged. On the final partition, we used hypergate to detect combinations of genes that best identify each cluster[84]. We then used these lists to train a classifier and assess its performance, taking an area under the curve above 0.7 as indication that clusters could be re-identified with a limited number of genes.

The approach with hypergate as well as regular differential gene expression analysis using MAST with nUMI as a variable to regress against yielded lists of markers for each clusters.

## Gene set enrichment analysis

GSEA requires ordered lists of genes rather than restricted lists of markers[85]. We used each gene to train a classifier to recover given clusters and used the area under the receiver-operator curve as the value for gene set enrichment analysis. The area under the curve is a bounded value between 1 and −1 and its sign indicates whether

increased expression of the gene improves recovery of the cluster (positive sign) or removal of the cluster (negative signe), thus its properties are particularly suitable to this type of analysis. We recovered gene sets from Zfin and carried out the analysis using the liger package https://cran.r-project.org/web/packages/liger/ (not to be mistaken with rliger which is dedicated to dataset integration[86]).

## Trajectory reconstruction

To identify step-wise transitions in expression from deep to shallow quiescence we performed a pseudo-ordering analysis with a custom algorithm inspired by a method developed by the Satija lab[76]. We first embedded cells in a neighborhood-preserving reduced space using diffusion maps on variable genes as implemented via the diffusionmap R package (https://cran.r-project.org/web/packages/diffusionMap/index.html). In this reduced space we computed a thousand minimum spanning trees using two thirds of the data and calculated the average distances between cells along bootstrapped minimum spanning trees. From this distance matrix we used multidimensional scaling to compute a three-dimensional embedding allowing good visualization of the reconstruction. In their implementation, Mayer et al. then finalized the reconstruction with a final minimum spanning tree[76]. However, previous studies showed that parametrizing a minimum spanning tree with a skeleton of nodes improves reconstruction and avoids short-circuits[87]. We thus used the ElPiGraph R package to fit a skeleton of up to a hundred nodes[88]. Noisy paths can form due to uneven density in the embedding, we thus pruned independent paths that emerged and terminated in the same cluster. We then completed the tree and projected each cell on its edges. Branches were then automatic.

## Synthesis of digoxigenin probes for in situ hybridizations

Total cDNA was synthesized from brain lysates through reverse transcription with Superscript II. Genes of interest were amplified via polymerase chain reaction with specific primer pairs derived from https://primer3.ut.ee/ and/or https://www.ncbi.nlm.nih.gov/tools/primer-blast/ and cloned with the help of the StrataClone PCR Cloning kit from Agilent. Genes that could not be cloned this way were ordered as gblock from IDT https://eu.idtdna.com/pages/products/genes-and-gene-fragments/double-stranded-dna-fragments/gblocks-gene-fragments with overhangs allowing restriction digest cloning into NEB10 bacteria (#C3019H). 5 μg of DNA was linearized with the appropriate enzyme, purified and transcribed in vitro in presence of digoxigenin-labeled uridine and precipitated with lithium chloride. Probes were then resuspended in 50 μL DNAse and RNAse-free water and stored at −80 °C.

*mfge8a* was cloned by PCR from cDNA using the following primer pair: CATCTGTGCCGAGGGATTTG (forward) TGATGATGCCTGTGA CCCTT (reverse). *vamp5* was cloned by PCR from cDNA using the following primer pair: GGCGCATTTAACCCACAGAT (forward) ACAC ATGCCAACCAAATCCT (reverse). *timp4.3* was ordered as a gBlock (IDT DNA) spanning nt 1-913 of ENSDART00000167824.3, flanked with T7/T3 RNA polymerase sites and a BamHI site. *slc4a4a* was ordered as a gBlock spanning nt 143-1336 of ENSDART00000124336.4, flanked with T7/T3 RNA polymerase sites and a BamHI site. Probes were directly transcribed in vitro using 1.6 μg gBlock DNA and T7 RNA polymerase.

## Preparation of tissue for immunohistochemistry and in situ hybridization

Whole brains or whole telencephala were dissected from 3 to 4 months old fish after euthanasia as described above in cold PBS. They were immediately placed in cold 4% PFA and fixed on a rotating platform at 4 °C overnight. The next day brains were dehydrated through sequential washes in 25%, 50%, 75% and finally 100% methanol (mixed with PBS the first three solutions) and then stored at −20 °C. For labeling, brains were first rehydrated through washes in 75%, 50% and 25% methanol in PBS and then processed for immunohistochemistry and/or in situ hybridization.

## Chromogenic in situ hybridizations

Adult brains were retrieved from -20 °C and rehydrated. They were subsequently treated with a bleaching solution made up of 0.5X SSC, 3% H2O2, 0.05% formamide diluted in DNAse and RNAse-free water. They were then washed 4 times in PBS + 0.1% Tween 20 and then pre-incubated at 65 °C in a solution containing 5X SSC, 65% formamide, 0.1% Tween 20, 50 μg/μL of heparin and 2.5 mM of citric acid for at least 4 h. The solution was removed and then added again with the labeled antisense probe for overnight incubation at 65 °C. The next day the probe was removed and the brains washed in increasingly stringent conditions up to 0.05X SSC to remove excess probe before being incubated in blocking buffer with an anti-digoxigenin antibody coupled to horseradish peroxidase (HRP) for 2 h. The brains were then washed overnight and the next day HRP activity was revealed with NBT/BCIP either on whole mounts or on 60 μm thick coronal slices prepared after the antibody step.

## Single molecule FISH with RNAScope

We sought to leverage the increased sensitivity of single molecule fish and its ease of coupling with immunostaining against proteins to better quantify the presence of specific cells and in particular to better distinguish between cells expressing high or low levels of *timp4.3*. The Hiplex probe for *timp4.3* (#ref 1167421-T7, consisting of a 13 ZZ probe targeting 173 - 984 of the sequence ENSDART00000187110.1) was ordered from Bio-Techne. The initial steps up to bleaching are the same as those performed for regular chromogenic ISH. Then, we pre-incubated brains at 40 °C either in RNAScope's proprietary diluent buffer from Bio-Techne (https://www.bio-techne.com/reagents/rnascope-ish-technology) or in chromogenic ISH buffer but with 25% formamide instead of 65%. Hybridization was performed in the same conditions as pre-incubation and we followed instruction from the Hiplex kit for the rest of the procedure.

## Immunohistochemistry

Adult brains were retrieved from −20 °C and rehydrated or processed after completion of ISH. Brains were kept in the blocking solution (5 % Normal Goat Serum, 0.1 % Triton-X, 0,1 % DMSO in PBS) (Sigma Life Science, 1002135493) for at least 1 h before incubating in the primary AB containing-blocking solution overnight at 4 °C. On the second day, brains were washed at least 4 times for 10 min with PBT and then incubated overnight at 4 °C in a blocking solution containing the secondary ABs. On the third day, brains were washed at least 3 times before imaging.

Primary antibodies:
-   Anti-ZO1: monoclonal, mouse, IgG1, 1:200, Thermo Fisher Scientific, Cat#33-9100
-   Anti-mCherry: Rabbit, 1:250, Takara #622496Secondary antibodies:
-   Anti-mouse IgG1: Goat IgG conjugated to Alexa Fluor 647, 1:500, Thermo Fisher Scientific, A-21240
-   Anti-rabbit IgG: Goat IgG conjugated to Alexa Fluor 546, 1:500, Thermo Fisher Scientific, A-11010

## Clonal analysis

Data for clonal analysis was reanalyzed from[23]. Traced fish were double transgenic generated from crossing Tg*(her4.1:*ERT2CreERT2*)/+*[89] and Tg*(-3.5ubb:*loxP-EGFP-loxP-mCherry*)/+*[90]. Irreversible expression of mCherry was sparsely induced by injections of low tamoxifen quantities at 3 months and fish were traced for up to 507 days. Cell types were identified on the basis of the expression of glial and proliferation markers as well as their position relative to the ventricle. For more information see[23]. Because glia and in particular astrocytes can proliferate in mammals, we considered all clones which included only RG,

whether one or many, as potentially originating from non-neurogenic cells and compared their proportion with that of cluster q4 cells. The proportion of q4 cells was estimated in the general population based on scRNAseq and on smFISH for *timp4.3*, which independently yielded very similar results with an estimate close to 50%. To determine whether cluster q4 cells were indeed induced by clonal recombination, *timp4.3* levels were quantified with smFISH in cells labeled with the recombined reporter gene (mCherry) and in randomly selected non-recombined cells in Dm at 6dpi, the earliest time point to detect mCherry. The area analyzed and the average number of clones induced per area (of the order of 20) were similar to the conditions in[23]. The proportion of cells expressing high levels of *timp4.3* as well as the overall distribution of *timp4.3* in the recombined versus non-recombined populations were compared.

### Re-analysis of published datasets

We downloaded raw matrices for around 40 datasets for re-analysis (see text for references). We subjected them to the same type of quality filtering and analysis as our own datasets without the consensus clustering approach. In most cases we were interested in synexpression of specific gene-sets independent of precise clustering. In the case of the Cebrian-Silla dataset[10] we used the gene sets identified in the paper to assign regional identity to the different clusters of neural stem cells. For the whole mouse telencephalon dataset[53] our own analysis agreed almost perfectly with the published annotations but we found that regional signatures from[10] were differentially enriched in subsets of the original "SVZRG" cluster and thus subdivided it further. Astrocytes were also heterogeneous but we could not assign a specific identity to astrocytic subsets and thus kept the broad annotations. Our analysis largely agreed with the originally published ones in almost all cases except for the one described in the main text[29].

Re-analyzed datasets are made available in a data.gouv repository (see Data Availability and Code availability). They were not included in this repository only if (1) they are already readily available through the original publication, (2) our analyses agree with that of the original publication or (3) we did not conduct analyses requiring further partition of the original data.

### Identification of orthologous genes lists

To compare lists of enriched genes between species and to perform direct mapping of data sets we needed reliable lists of orthologs. We found that commonly used methods limit the feasibility of this type of comparison. A widespread approach is to convert all of the genes in a dataset to their orthologs through Ensembl (https://www.ensembl.org/index.html) and to then take all one to one orthologs for integrated analysis. We found that annotation through Ensembl makes many mistakes, both false positive and false negative. Moreover because teleosts underwent an additional round of whole genome duplication compared to other vertebrates, many mammalian genes map to multiple orthologs in zebrafish which leads to them being discarded from such analyses. We thus curated a lists of orthologs based on a common syntax originating from the one used by eggnog[91]. We identified orthologs for zebrafish genes by integrating information from eggNOG, Alliance Genome (https://www.alliancegenome.org/), Ensembl and Zfin (https://zfin.org/) both through command line tools and manual curation of identified relationships (Supplementary Dataset 1). We mapped mouse genes to the same set of orthologs in a similar manner but without input from Zfin (Supplementary Dataset 2). The common syntax being based around eggNOG's output then allowed us to directly compare genes with orthologs mapped through eggNOG for other datasets for example in birds or lizards. We also mapped genes from less studied species through eggNOG and Ensembl Metazoa to assess the expression of individual genes in those datasets.

### Attribution of astroglia, ribosomal and neuroblast scores

In order to identify cell populations of interest, in particular paRG, and to summarily illustrate some of our conclusions we used the Seurat scoring method with gene sets of interest. This scoring method identifies "background genes" among genes with a similar average of expression and for each gene belonging to a list of gene of interest. It then scores each cell for the relative expression of the genes of interest compared to that of the background genes, highlighting cells with a high level of expression for a set of genes that cannot be explained only by the average level of expression of those genes in the dataset. Importantly, this means that the scores reflect relative and not absolute levels of expression, thus a set of genes with a high level of expression uniformly distributed across all cells will not yield high scores.

We chose *GFAP*, *GLUL*, *CX43/GJA1* and FABP7 to compute a glial score to label astroglial cells based on their widespread use as markers of astroglia across many species. We chose *NEUROD1*, *NEUROD2*, *NEUROD4*, *NEUROD6*, *STMN1*, *DPYSL3*, *GAP43*, *BHLHE22* and *TUBB5* to compute a neuroblast score. Finally, we computed a ribosomal score based on all ribosomal protein genes expressed in each dataset, in order to highlight stem cells that are close to proliferating among stem cells that do not belong to the proliferating cells clusters. Indeed, it has been shown in several previous studies that stem cells, including adult NSCs[87], upregulate ribosomal protein genes on their way to activation, presumably to cope with the metabolic demands associated with mitosis. High scores for ribosomal genes expression are also found in proliferating cells and in early post-mitotic neuroblasts as expected given the need for them to establish their differentiated cell morphology after commitment.

### Cross-species mapping

To map our zebrafish data to the one generated by[53] we used our curated lists of orthologs for both mouse and zebrafish, generated matrices with the same gene names and aggregated them based on genes that were expressed in both datasets. We identified variable genes using the variance stabilizing transformation[92] and used those as input to MetaNeighbour[57] to unbiasedly identify similar cell types between the two species.

Several groups have now developed tools to perform integration of multiple datasets followed by a joint analysis. Among them, SAMap[93] is particularly suited to integrate data from multiple species. However, the main advantage of such analyses lies in their ability to systematically integrate data over many different cell types simultaneously. Moreover, parameter optimization can be difficult and somewhat subjective and thus possibly introduce a differential bias in the analysis. Finally, the use of integration methods requires finding a balance between compensating for inter-dataset variance in a meaningful way without excessively masking intra-dataset variance. In our case, we focused on a restricted number of cell types which meant that iterative analysis of several datasets remained tractable. Moreover, the datasets we used varied substantially in terms of methodology used, sequencing depth and level of heterogeneity. Therefore, to avoid any potential confounding effect introduced by integration and because analysis of each dataset was still possible in our case, we chose not to use any integration method for our comparative analyses.

### Reporting summary

Further information on research design is available in the Nature Portfolio Reporting Summary linked to this article.

## Data availability

All sequencing data generated in this study have been deposited in the GEO database and are available at: https://www.ncbi.nlm.nih.gov/geo/query.cgi?acc=GSE225863. Additional code and processed data have been deposited in the French data.gouv database and are

available at: https://entrepot.recherche.data.gouv.fr/privateurl.xhtml?token=bede1d62-f7cf-4e85-b6ce-05121176f108. Source data for all plots generated with data other than those already made available through the repository sequencing data are provided with this paper (see Source Data File). Source data are provided with this paper.

## Code availability

The data.gouv repository mentioned above includes an R file with custom functions and rds files for our dataset as well as most of the re-analyzed datasets. Although most of the analyses were conducted using Seurat v2.3.4 to store the data, we provide them in list format with accompanying custom plotting functions so that they can be downloaded and used with minimal coding experience and independently of the specific local R installation. We also provide files to directly query the expression of genes relevant in our analyses, in particular the reconstruction of gene expression along the neurogenic cascade and the identification of cells expressing genes belonging to the astrocytic geneset, across the different datasets. Finally, the site includes a README file detailing the different datasets there, how to download them and how to extract raw or normalized data and/or plot genes of interest.

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

## Acknowledgements

We thank the ZEN team for input, the Institut Pasteur CB UTechS service platform for expert assistance with FACS analysis and scRNAseq processing, and Thibaut Brunet for critical reading of the manuscript and valuable advice. We are greatly indebted to Isabel Fariñas and Barbara Treutlein for sharing data prior to publication, and for insightful discussions. Figure panel 1a was made in BioRender.com. Work in the L. B-C. laboratory was funded by the ANR (Labex Revive), La Ligue Nationale Contre le Cancer (LNCC EL2019 BALLY-CUIF), the Fondation pour la Recherche Médicale (EQU202203014636), CNRS, Institut Pasteur and the European Research Council (ERC) (SyG 101071786 – PEPS). D.M. is a member of the Médecine Science MD-PhD program and Sorbonne Université and was recipient of a PhD student fellowship from Ecole Normale Supérieure.

## Author contributions

Conceptualization: DM, LBC; Methodology: DM, IF, AA; Funding acquisition: LBC; Project administration: LBC; Supervision: LBC; Writing – original draft: DM; Writing – review & editing: DM, LBC.

## Competing interests

The authors declare no competing interests.
