## [Peer Review File · Nature Communications]

Reconstruction of macroglia and adult neurogenesis evolution through cross-species single-cell transcriptomic analysesREVIEWER COMMENTS

Reviewer #1 (Remarks to the Author):

In the manuscript "Integrative single-cell transcriptomics clarifies adult neurogenesis and microglia evolution", Morizet and colleagues analyze single-cell RNA sequencing data in zebrafish and other vertebrate and invertebrate species to clarify the evolution of macroglia cells.

The authors present a new scRNAseq dataset from the telencephalon of adult zebrafish, describe the diversity of GABAergic neurons, and then focus on glial cells and on their similarities and differences with glia from other species. Their data and analysis support the hypothesis that mammalian astrocytes evolved from astrocyte-like radial glia in vertebrate ancestors by subfunctionalization. Although similar ideas were already present in the literature (see for example Laywell et al PNAS 2000 PMID: 11095732), this paper presents for the first time a comprehensive analysis of single-cell transcriptomic data focused on glia evolution. This work will be of great interest to researchers in the fields of evolutionary neuroscience and adult neurogenesis.

The analysis is appropriate for the study's objectives and with choices justified convincingly in the methods section.

I hope that the suggestions below will help the authors strengthen the manuscript and clarify the key points:

1. Radial glia

1.1. Evolution of astrocytes in vertebrates

The authors propose that mammalian radial glia and astrocytes are sister cell types, that evolved by subfunctionalization of an ancestral astrocyte-like radial glia cell. As proposed by Gunter Wagner, new cell types may originate by genetic individuation of distinct cell states, for example a homeostatic and a stress-induced state. Here, the clonal analysis data indicate nicely that zebrafish radial glia transition between an astrocyte-like state and a neurogenic state, further supporting the evolutionary diversification scenario.

In my opinion, this is the most important message of the paper.

1.1.1. Can alternative hypotheses be ruled out more convincingly? For example, a potential alternative hypothesis is that non-mammalian astroglia are "just" like mammalian radial glia that did not complete their differentiation into astrocytes. This view would build on the naive and widespread assumption that the brains of non-mammals are "less mature" than mammalian brains. The ensuing prediction is that mammalian adult neural stem cells would resemble more the ancestral radial glia, and that mammalian astrocytes are a further step ahead in the differentiation cascade.

In contrast, the cell type diversification scenario predicts that neural stem cells in the SVZ and DG should not express astrocytic markers. Is this the case? This should be easy to test.

1.1.2. I understand the practicality of describing and analyzing clusters of single-cells, but in this case, "clusters" q2 to q4 seem to be cell states of the same radial glia cells. Shouldn't these cell states be described as extremes in a continuum? How separable are these clusters, and what is the biological interpretation of their separability?

1.1.3. Are there, by any chance, any transcription factors expressed only in mammalian astrocytes but not expressed in astrocyte-like radial glia? It would be interesting to gain some insights into the molecular changes associated with the genetic individuation of astrocytes in the mammalian lineage.

1.2. q6 cluster

Q6 coexpresses *gsx2* and the pallial marker *emx2*. This is surprising. Is this comparable to the mammalian lateral ventricle, mentioned in the text? (Line 96) Are there other "subpallial" markers expressed in q6, such as *dlx* genes (figure 2e)? It is important since the authors mention that these cells were previously thought to be pallial (line 105)

1.3. Macroglia in invertebrates

1.3.1. Among the many invertebrate species surveyed by the authors, I do not see amphioxus. Considering that ascidians may have undergone secondary simplification of parts of their nervous system, cephalochordates are valuable to make more precise inferences on chordate ancestors. Bozzo et al PMID: 33624886 and Ma et al PMID: 35732129 might be useful references to explore this further.

1.3.2. the *C.elegans* dataset used in this analysis is not the most up-to-date. It might be interesting to confirm the original findings by using the CENgen dataset <https://www.cengen.org/single-cell-rna-seq/>

2. GABAergic neurons

2.1. Was there any specific indication from previous literature that MGE and LGE exist in zebrafish (or in fish in general)? This should be stated clearly in the text to clarify the novelty of this manuscript (besides the identification of putative CGE-derived neurons).

2.2. When describing the GABAergic subpopulations in the main text, it would be helpful to refer to the cluster numbers in figure 1c.

2.3. not all telencephalic GABAergic neurons are interneurons (many are projection neurons!). Please correct line 272 and others where the two terms are being conflated.

3. Description of experimental procedures and results in the main text:

3.1. the main text needs more details, precision, and quantitative statements. For key analyses and experiments, readers should not have to go to the methods to figure out what has been done.

Examples:

-- line 59: mention FACS sorting;

-- line 114: how was the re-analysis done? Figure 3 is the only figure provided to illustrate the comparison of glial cells across species, and it is unclear how the labels "qRG", "paRG" and "NB" were assigned. This needs to be clarified in the main text and/or in the methods. Figure 3 could be expanded to show some examples/quantifications from the primary data.

- line 160: "Among the genes associated with deeper quiescence in zebrafish RG, several of them were expressed in both qRG and astrocytes in mice". How many is "several"?

- line 168: mention how the mapping of zebrafish glia to mammalian data was done.

3.2. some statements in abstract and main text seem exaggerated and unsupported by the data; examples

-- line 60: "full extent of cell diversity"; the dataset is small and does not capture the full diversity of glutamatergic and GABAergic neurons. I could not find anywhere a statement on how many cells were sequenced from zebrafish.

-- "comprehensive" in line 269 also seems an overstatement (if it refers to neurons)

3.3. Definitions

Providing exact definitions for the terms used would be useful. Examples: what is a "pre-activated RG"? (line 120); what definition of "astrocyte" are the authors following? (line 164 states that besides mammals most species are thought not to have astrocytes, but other scientists had referred to

zebrafish glial cells as astrocytes...)

3.4. Content and clarity of the main figures

- figure 1b: Add axes to the figure to indicate this is a tSNE
 - the labels of the ancestors in figure 3 are confusing
 - indicate cluster number in figure S4
 - figure 5: at the very least, add a figure legend to indicate what the colors mean (the information is in the caption, but having a legend in the figure itself makes it easier to read).
- In general, I thought that the figures include too many summary diagrams and that all the interesting data is in the supplementary file. The authors may want to consider moving some of the supplementary figures, in particular figures 18 and 19, to the main figures.

Reviewer #2 (Remarks to the Author):

In the manuscript entitled "Integrative single-cell transcriptomics clarifies adult neurogenesis and macroglia evolution", Morizet et al. used zebrafish model and a very comprehensive meta-analysis of transcriptomic data across a wide range of animal species to address the evolutionary changes in neuronal and glial cell types, in search for homologies and differences.

Overall, the authors conducted a thorough and meticulous computational analysis of their single cell transcriptomic dataset, as well as they reanalyzed previously published datasets. Multiple control analyses were performed to ensure the accuracy of clustering. Utilizing their computational workflow, the authors were able to uncover novel insights from published data and examine the topic with an unparalleled level of resolution. The text is solid, concisely and logically written, technically sound, and most conclusions/interpretations justified by the results obtained. The results of this work will surely increase our current knowledge of neural cell type (both glial and neuronal) evolutionary aspects and heterogeneity. In my view, it could be published with some minor revisions.

Here, below, I have some advice:

- The Authors propose/show that features of neural progenitors, especially neurogenic processes, can be highly conserved through evolution (Author's text: "ventricular progenitor patterning is maintained not only throughout life but also throughout evolution"). Then, in supplementary material they address the remarkable differences existing among orders/species, as well as among mammals, especially referring to the remarkable decrease of adult neurogenesis in humans (generally large-brained species) with respect to laboratory rodents.

I think that both these statements are true, I'm just wondering whether the reader might make confusion between the well conserved features of neurogenesis (both in ontogeny and in evolution) and the differences that occur among species. Maybe this should be made clearer in the main text (not only in supplementary), by underlying such difference. I say this because, in the current controversy whether human hippocampal neurogenesis persists or not in the adult, many people confuse (or use) the evolutionary conservation of the processes/progenitors/genes to support the general notion that existence/persistence of adult neurogenesis in humans can be true or logically acceptable.

In other words: the fact that neurogenic processes are conserved through evolution does not mean that their occurrence, location, and rate must be the same in different species.

Minor points:

- SEZ is an old terminology, usually replaced by SVZ

Supplementary information

- Lines 20 and following (paragraph: Homology of neurogenic niches)
For general readers maybe it's better to remind/explain the homologies/differences occurring in the regions of the olfactory system and hippocampus between zebrafish and mammals

- Line 73-76: Author's text: "...which will shed light on the origin of cell type diversification in the hippocampus. This is of particular interest for the study of adult neurogenesis as the dentate gyrus niche has been described as being more of a mammalian innovation²³ "

This thesis by Kempermann was considered wrong by evolutionists (see Commentary by Powers A Brain Behav Evol. 2013, on another review article by Kempermann, but on a similar concept)

- Lines 89-91: Author's text: "... Thus, although there is some evidence that immature neurons can still be detected in adult humans, adult neural stem cells remain elusive, and it is more largely agreed that they are likely prematurely depleted compared to most other species ³⁷"

Maybe the concept of "immature" neurons as I think it is intended here (non-newlyborn, "dormant" neurons, such as those described in the cortex layer II) should be explained to the reader. The current, provisional word "immature neuron" can be confused with the newlyborn neurons produced from active stem cell niches in adult neurogenesis; they also share a phase of immaturity with expression of the same markers.

- Lines 109-115: Author's text: "...Contrary to arguments explaining the low neurogenesis in humans based on a selection against plasticity due to the size and complexity of their brain⁴⁶, this interpretation suggests a tradeoff with an expanded neocortex."

I do not think that this is "contrary". The brain size and neocortex expansion are usually correlated. And this does not exclude the tradeoff with expanded neocortex.

Author's text: "Selection for these traits would have taken place at a moment when sight was already the primary sensory modality thus reducing selective pressure on olfactory processing⁴⁷, and when life expectancy was much shorter, thus resulting in only a short period of time without hippocampal neurogenesis given that it persists well into teenage years."

This is also supported by work of Aboitiz & Montiel on the origin of isocortex.

- Line 170 and following: Author's text: "we scored mouse astroglial cells for genes enriched in q4 over q2"

The authors compare the cells in the q4 and q2 clusters to identify markers that resemble the transcriptional signature of astrocytes. However, it is not entirely clear why they selected these two clusters over the other dorsal qRG clusters (q1, q3). The authors should provide a better explanation and justification for their selection.

- The authors frequently refer to the expression levels of specific genes across and within different species in the manuscript. However, they often do not present absolute expression levels in the form of plots or tables. Instead, they provide representations of relative expression levels or schematic summaries (e.g., Figure 3, Figure 5, etc.). To support their discussion, the authors should include through the whole manuscript evidence of the expression levels of the genes they refer to in the form of tables with raw data or expression plots.

- Line 179 and following: The authors should provide a more comprehensive explanation for the clonal analysis and clarify in the main text that the labeling of Her4 cells will not be exclusive to q4 cells but will also include cells from other clusters. This is apparent from the legend of Extended Data Figure 14, but it is not adequately explained in the main text. Additionally, it is unclear from the text when

the proportion of q4 cells is estimated. Based on the text, it appears to be at 507 days of chase, but it should instead coincide with the time of Tx administration. Moreover, how can the authors conclude that the difference is not due to cell death?

- Line 236: please, provide original data in table or plot for this statement.

- Extended data figure 3: it is not clear here how the two plots should be compared. The plots here should be more informative to make the comparison clearer also for non-experts.

- Extended data figure 6: Clusters classification should be indicated in the plots to simplify the understanding of the plot, similarly to figure 17

- One notable advantage of this study compared to others is the final number of cells included in the analysis after all filtering steps. Therefore, it would be beneficial to clearly state this number in the main text

Summary chart of reviewers' comments

(Main changes are highlighted in grey in the revised text, legends and supplementary data)

	Revision type	Data or info already acquired	Status
Reviewer 1			
1.1.1. Counter hypothesis: could zebrafish RG be simply "immature astrocytes"?	In silico analyses of 2 additional datasets: - zebrafish juvenile - mouse embryo	✓ Hypothesis likely ruled out: - changes between juvenila and adult RG in zebrafish - early specification of astrocytes in mouse - specification of ependymal cells occurs in non-mammals	New Fig. S14 + Main text l.194-200.
1.1.1. Further support to proposed hypothesis: mouse RGL should not express astrocytic markers	None, was already shown (Fig.S11-S13)	✓ Hypothesis confirmed	NR
1.1.2. Clusters q2 and q4 are more of a continuum	None, we made sure that our text is not taking strong position	✓ We agree, this is reflected by the scenarios proposed in Fig.S15 (scenarios F & G)	NR
1.1.3. Transcription factors specific of astrocyte vs RGL in mammals?	In silico analysis	✓ Search done. We did not find any.	Response to referee
1.2. Cluster q6 expresses both gsx2 and emx2 and was thought to be pallial (Cosacak 2019)	In silico analyses	✓ Conclusion confirmed: - Emx2 of broad expression in mouse SVZ as well, - Cosacak et al. show the correct expression profile although conclude wrongly,	Response to referee, figure provided for referee (Fig.R1.1)
1.3.1. Add Amphioxus data (2 papers suggested)	In silico analysis on second paper (1 st paper does not contain data)	✓ Done: Conclusions apparently confirmed (but low discrimination power of the data)	Main text l.246-251
1.3.2. Use up-to-date C. elegans data. Use CENgen dataset.	In silico analysis	✓ Done: Conclusions confirmed	Ref added (ref. n°61)
2.1. Provide refs for previous identification of zebrafish MGE and LGE	Text	✓ Done	Text l.71-73 and ref. n°24
2.2-2.3. Various corrections to text	Text	✓ Done	Text
3.1. Add experimental details to text (e.g., l.59;114;160;168)	Text, methods, figures	✓ Done	Main text l.64-66, 122-124, 181-183
3.2. Delete overstatements (l.60;269)	Text	✓ Done	Text
3.3. Definitions of pre-activated RG and astrocytes	Text	✓ Done	Text l.128-130 and l.37-38
3.4. Content and display of figures: add "real" data to main Figures	Figures and legends	✓ Done	Figs.3 and 4 extensively modified

Reviewer 2			
Main: make the difference clear between evolutionary conservation and occurrence in individual species	Text	✓ Done	Text
Minor: replace SEZ with SVZ	Text	✓ Done	Text
Supp. l.20: insufficient info on zf pallium	Text	✓ Done: added a descriptive § on pallial subdivisions in zf	Supp. Text l.54-55 and l.66
Supp. l.73-76: G. Kempermann's hypothesis not always accepted	Text	✓ Done: we agree, and phrased our text better	Supp. Text l.83-86
Supp. l.89-91: clarify "immature neurons »	Text	✓ Done : we agree	Supp. Text l.99-103
Supp. l.109-115 : - clarify selection and trade-offs in human brain evolution - mention Aboitiz and Montiel	Text	✓ Done	Supp. Text l.123-126 and ref.54
Supp. l.170: justify use of q2 and q4	Text	✓ Done	Main text l.186-189
Add "real" data information to results using gene expression	Figures and legends	✓ Done (see also Referee 1 point 3.4)	More plots available in online repository; Fig.3 extensively modified
Main l.179: provide time point zero for her4:ERT2CreERT2 tracing	Experimental, time point 0 + ISH with q4 marker	✓ Done	New. Fig.4e,f, new Fig.S15
Main l.236: provide data for sox9b and oligos	In silico analysis	✓ Done	Plots in data.gouv repository
Supp. Fig.3: provide explanations	Text	✓ Done	Fig. legend
Supp. Fig.6: provide explanations	Text	✓ Done	Additions to plot in Fig.S4
Indicate cell numbers	Text	✓ Done	Text l.52-54

Itemized response to our Referees

Referee 1

In the manuscript "Integrative single-cell transcriptomics clarifies adult neurogenesis and microglia evolution", Morizet and colleagues analyze single-cell RNA sequencing data in zebrafish and other vertebrate and invertebrate species to clarify the evolution of macroglia cells.

The authors present a new scRNAseq dataset from the telencephalon of adult zebrafish, describe the diversity of GABAergic neurons, and then focus on glial cells and on their similarities and differences with glia from other species. Their data and analysis support the hypothesis that mammalian astrocytes evolved from astrocyte-like radial glia in vertebrate ancestors by subfunctionalization. Although similar ideas were already present in the literature (see for example Laywell et al PNAS 2000 PMID: 11095732), this paper presents for the first time a comprehensive analysis of single-cell transcriptomic data

focused on glia evolution. This work will be of great interest to researchers in the fields of evolutionary neuroscience and adult neurogenesis.

The analysis is appropriate for the study's objectives and with choices justified convincingly in the methods section.

I hope that the suggestions below will help the authors strengthen the manuscript and clarify the key points:

1. Radial glia

1.1. Evolution of astrocytes in vertebrates

The authors propose that mammalian radial glia and astrocytes are sister cell types, that evolved by subfunctionalization of an ancestral astrocyte-like radial glia cell. As proposed by Gunter Wagner, new cell types may originate by genetic individuation of distinct cell states, for example a homeostatic and a stress-induced state. Here, the clonal analysis data indicate nicely that zebrafish radial glia transition between an astrocyte-like state and a neurogenic state, further supporting the evolutionary diversification scenario.

In my opinion, this is the most important message of the paper.

1.1.1a. Can alternative hypotheses be ruled out more convincingly? For example, a potential alternative hypothesis is that non-mammalian astroglia are "just" like mammalian radial glia that did not complete their differentiation into astrocytes. This view would build on the naive and widespread assumption that the brains of non-mammals are "less mature" than mammalian brains. The ensuing prediction is that mammalian adult neural stem cells would resemble more the ancestral radial glia, and that mammalian astrocytes are a further step ahead in the differentiation cascade.

Whether mammalian astrocytes are cells that develop "differently" or cells that develop "more" than astroglial cells in non-mammalian species which do not possess astrocytes is indeed a key question related to our work and we thank the reviewer for pointing out this issue. Although this is difficult to rigorously address experimentally, we believe that converging lines of evidence argue against the hypothesis that astrocyte-like radial glia in non-mammalian brains are merely astrocytes with a stunted development :

- 1) In zebrafish, the expression of the genes that belong to the astrocytic geneset (see Table S1, and new Fig.S14a) is higher in adults (3mpf, present dataset) than in post-larval juveniles (6dpf, 15dpf, Pandey et al., *Genome Res.*, 2023). This suggests that the maturation of astrocyte-like radial glia is a gradual process and that their presence in the adult brain is not a form of neoteny.
- 2) In mice, glioblasts generated during the late embryonic stages, and that delaminate from the ventricular zone to give rise to parenchymal astrocytes, already express genes of the astrocytic geneset that set them apart from radial glia (see plots generated from <http://mousebrain.org/wheel/>, new Fig.S14b). There are two implications to this observation: first, expression of the astrocytic geneset is an early trait during development of the mouse brain; second, the delamination process and the upregulation of astrocytic genes happen concomitantly, and there does not exist an extended period during which two different populations of radial glia co-exist at the ventricular zone, one positive for astrocytic markers and another one negative and that would delaminate and give rise to astrocytes. Thus, instead of being an additional step added to the maturation process of

astrocytic-like radial glia from non-mammalian species, the delamination of astrocytes instead happens during the early stages of maturation.

- 3) Multiciliated ependymocytes can be found in the brains of reptiles that do not have parenchymal astrocytes (cf our re-analyzed dataset of *Pogona vitticeps*). In mammals, ependymocytes are the last cells to be generated from developmental radial glia in the early postnatal stages, after astrocytes have already been produced. This suggests that later steps of the differentiation and maturation process of radial glia do happen in non-mammalian species, thus that parenchymal astrocyte generation is not a phenomenon that is added on top of the developmental process of non-mammalian species.

Altogether, these observations show that astrocyte-like radial glia mature over a long time, that the delamination of astrocyte progenitors from developmental radial glia happens roughly at the same time as astrocytic genes are turned on in mammals, and that radial glia in non-mammalian species can give rise to cell types that are generated later than astrocytes in mammals. Thus, the production of parenchymal astrocytes in mammals likely happens early during differentiation of the radial glia lineage, rather than being a late step, added after radial glia have reached the level of maturation that they display in non-mammals.

→ Some of these arguments have now been added to the text (Main text l.194-200) and are illustrated in New Fig.S14.

1.1.1b. In contrast, the cell type diversification scenario predicts that neural stem cells in the SVZ and DG should not express astrocytic markers. Is this the case? This should be easy to test.

This is indeed the case, although not to the level of a perfect binary dichotomy between genes expressed or absent in astrocytes vs NSCs but rather in terms of noticeable differences in the level of expression. Table S1 and Fig.S11, S12, S13 show examples of such genes, enriched in q4 (when compared to q2) in zebrafish and in astrocytes (when compared to NSCs) in mice in both SVZ and SGZ.

1.1.2. I understand the practicality of describing and analyzing clusters of single-cells, but in this case, "clusters" q2 to q4 seem to be cell states of the same radial glia cells. Shouldn't these cell states be described as extremes in a continuum? How separable are these clusters, and what is the biological interpretation of their separability?

Indeed the nature of the relationship between q2 and q4 is not entirely clear even to us. The clusters can be re-identified with a good accuracy using a simple classifier trained on a handful of marker genes (see Fig.S2b and S2c) suggesting good separability. However, the transitions in levels of expression for individual genes appear to be gradual instead of there being sharp boundaries between the two clusters (for example see Figs.2b and S9). Moreover, previous studies from our lab have shown the existence of distinct intermingled populations of qRG in the zebrafish pallium (for example, Than-Trong et al., Sci. Adv., 2021; Mancini et al., Sci. Adv., 2023), but it is also largely accepted that qRG exist in a continuum of states of different quiescent depths. Because of this, besides using the clustering to identify differentially expressed genes, we tried to avoid stating that they correspond either to distinct subpopulations or to substates on a continuum (and in fact, our hypotheses depicted in Fig.S15 - hypotheses F and G- do cover both scenarios). We believe that scRNA-seq is not sufficient to distinguish between the two (and in fact questions about whether reported clusters correspond to distinct subpopulations or substates arise frequently, even in studies that do not focus on developmental processes which are particularly at risk). We have started trying to integrate labelling with cluster markers in situ and other assays to get a comprehensive picture of qRG dynamics and the precise relationships between subgroups defined by our markers. However, we expect this to require a few years and technical efforts before we can conclusively answer this question in a rigorous way.

1.1.3. Are there, by any chance, any transcription factors expressed only in mammalian astrocytes but not expressed in astrocyte-like radial glia? It would be interesting to gain some insights into the molecular changes associated with the genetic individuation of astrocytes in the mammalian lineage.

We surveyed different regions of the murine telencephalon (regional heterogeneity having been well described among astrocytes there) but did not find any transcription factor detected in astrocytes while not being expressed in zebrafish RG. We therefore did not change our text on this aspect.

However, for information, we noticed that some of the activated RG in adult mammalian brains express *Zeb1*, which is not expressed in species that do not have parenchymal astrocytes. *Zeb1* is expressed early on in glioblasts (Zeisel et al., Cell, 2018), and has been shown necessary and sufficient to prevent astrogenesis (Gupto et al., Cell Rep., 2021). Given that *Zeb1* drives the Epithelial-to-Mesenchymal transition, one hypothesis regarding the generation of astrocytes from RG is that, in mammals, *Zeb1* can be turned on and promote their delamination, without necessarily remaining expressed later. This being purely theoretical at this point, we have preferred not to modify our text.

1.2. q6 cluster: q6 coexpresses *gsx2* and the pallial marker *emx2*. This is surprising. Is this comparable to the mammalian lateral ventricle, mentioned in the text? (Line 96) Are there other "subpallial" markers expressed in q6, such as *dlx* genes (figure 2e)? It is important since the authors mention that these cells were previously thought to be pallial (line 105)

This is indeed important. However, several arguments support our interpretation on the location of q6 RGs:

- In the adult mammalian SVZ, *Emx2* expression seems to be largely unrestricted by region contrary to what would be expected (see below Fig.R1.1A-D, illustrating this in cells from Cebrian-Silla et al., 2021).
- In zebrafish, although *dlx2a* and *dlx2b* are detected in populations of inhibitory neurons, they are expressed at very low level if at all in RG, whereas *Dlx2* expression is also high in lateral wall SVZ in the adult mouse brain (Fig.R1.1E). This difference could be linked to a difference in sensitivity. Indeed, in the mouse dataset, although *Dlx2* is detected in RGL, it is expressed at much higher levels in their progeny cells (Fig.R1.1F,G). If the same is true in zebrafish, the expression of *dlx2a* and *dlx2b* might not be high enough to be detected in RG.
- important arguments for q6 being regional and located close to the pallial-subpallial boundary come from ontogeny and neurogenesis properties. First, the location of *gsx2*-positive RG in zebrafish (Fig.2C) matches the one predicted to be homologous to the mouse LGE in the prevalent eversion model (Folgueira et al., Neural Development, 2012). Second, this region displays higher proliferation levels than more dorsal pallial areas, is surrounded by inhibitory neurons, and seems to feed new neurons into a rostral migratory stream homolog (e.g., Adolf et al., Dev. Biol. 2006). These properties are reminiscent of the mouse lateral SVZ.
- finally, the in situ hybridization provided in Cosacak et al., 2019 (which proposes that *gsx2*+ cells are RG from the dorsal pallium) is in fact consistent with our interpretation (see top left panel of Figure 2d in this paper, <https://www.sciencedirect.com/science/article/pii/S2211124719304292>).

Based on this whole series of arguments, we have not amended our initial text.

Figure R1.1. **Expression of regional markers in the mouse SVZ**

A, UMAP plot restricted to Radial Glia like cells from the SVZ (retrieved from Cebrian-Silla et al). **B-E**, expression of the developmental regional markers *Emx1*, *Emx2*, *Gsx2* and *Dlx2* in adult RG from the SVZ. Although *Emx1* and *Gsx2* remain restricted to distinct subpopulations of RG, *Emx2* expression becomes broader postnatally. **F**, tSNE plot of all cells included in the SVZ dataset from Cebrian-Silla et al. **G**, expression of *Dlx2* projected on all cells, highlighting that its levels of expression are much higher in committed cells than in RG.

1.3. Macroglia in invertebrates

1.3.1. Among the many invertebrate species surveyed by the authors, I do not see amphioxus. Considering that ascidians may have undergone secondary simplification of parts of their nervous system, cephalochordates are valuable to make more precise inferences on chordate ancestors. Bozzo et al PMID: 33624886 and Ma et al PMID: 35732129 might be useful references to explore this further.

Indeed we were very interested in including amphioxus in the study. The publication by Bozzo et al. consists in ISH with a handful of markers including GS but is not comprehensive enough to be relevant here, in particular because all the genes of our astrocytic geneset have not been tested. The data also cannot be easily exploited for a reanalysis. We however contacted the authors of Ma et al when their paper came out to ask for access to the adult data (the online database accompanying the paper only included data from larvae). In this adult dataset, which the authors kindly shared, we were not able to find glia expressing the gene markers that we identified from the other comparisons. However this could be due to the very low library complexity of the cells from this study: since the aim was more focused on a first large-scale characterization, the authors used a shallow sequencing depth. Thus, the number of genes/cell is very low and the diversity of detected genes is also very low, which does not allow us to ensure that when a gene is not detected in a putative cluster of interest it is at least detected in other cells from the dataset. In these conditions, we cannot discriminate between amphioxus having glia that are different from those observed in vertebrates, or the study not having the power required to perform these comparisons. We added a comment on this in the text 1.246-251.

1.3.2. the *C.elegans* dataset used in this analysis is not the most up-to-date. It might be interesting to confirm the original findings by using the CENgen dataset <https://www.cengen.org/single-cell-rna-seq/>

We had indeed missed this dataset and we are grateful to the reviewer for pointing it out. We have now also queried this dataset for our genes of interest while using the cell populations identified in the

original papers and we were not able to find any glial cluster enriched for our astrocytic gene signature, supporting our previous conclusions. We now added this reference to the text (ref. n°61).

2. GABAergic neurons

2.1. Was there any specific indication from previous literature that MGE and LGE exist in zebrafish (or in fish in general)? This should be stated clearly in the text to clarify the novelty of this manuscript (besides the identification of putative CGE-derived neurons).

A previous study has indeed already looked at the expression of regional markers derived from mouse in the zebrafish telencephalon and come up with subdivision models. We have now included a more direct mention of this work (ref. 24, Ganz et al., 2012) to make it clear that the existence of LGE and MGE homologs was already supported before our study (text l.71-73).

2.2. When describing the GABAergic subpopulations in the main text, it would be helpful to refer to the cluster numbers in figure 1c.

We added the names of the clusters we were referring to in the text (l.74-79).

2.3. not all telencephalic GABAergic neurons are interneurons (many are projection neurons!). Please correct line 272 and others where the two terms are being conflated.

We apologize for this mistake and thank the reviewer for pointing it out. This has now been corrected (Discussion l. 307).

3. Description of experimental procedures and results in the main text:

3.1. the main text needs more details, precision, and quantitative statements. For key analyses and experiments, readers should not have to go to the methods to figure out what has been done. Examples:

-- line 59: mention FACS sorting;

We added a short description of how NSCs were enriched, mentioning cell-sorting on l.62-66.

-- line 114: how was the re-analysis done? Figure 3 is the only figure provided to illustrate the comparison of glial cells across species, and it is unclear how the labels "qRG", "paRG" and "NB" were assigned. This needs to be clarified in the main text and/or in the methods. Figure 3 could be expanded to show some examples/quantifications from the primary data.

- We have now added information on how the analysis was done (l.121-126) as well as an example Rmd file on our public repository accompanying the publication (<https://entrepot.recherche.data.gouv.fr/privateurl.xhtml?token=bede1d62-f7cf-4e85-b6ce-05121176f108>). We more directly explain what our repository contains in the manuscript (Main text l.633-645): *"This site includes rds files for our dataset as well as most of the re-analyzed datasets unless. Re-analyzed datasets were not included only if 1) they are already readily available through the original publication, 2) our analyses agree with that of the original publication and 3) we did not conduct analyses requiring further partition of the original data. Although most of the analyses were conducted using Seurat v2.3.4 to store the data, we provide them in list format with accompanying custom plotting functions so that they can be downloaded and used with minimal coding experience and independently of the specific local R installation. We also provide html file to directly query the expression of genes relevant in our analyses, in particular the reconstruction of gene expression along the neurogenic cascade and the identification of cells expressing genes belonging to the astrocytic geneset, across the different datasets. Finally, the site includes a README file detailing the different datasets there, how to download them and how to extract raw or normalized data and/or plot genes of interest"*.

- We have also added explanations on how the different cell types were identified (text l.126-130 and Materials and Methods l.588-608, "Attribution of astroglia, ribosomal and neuroblast scores").

- Finally, we added some illustrations in an extensively revised Figure 3.

- line 160: "Among the genes associated with deeper quiescence in zebrafish RG, several of them were expressed in both qRG and astrocytes in mice". How many is "several"?

It is difficult to give a precise quantitative answer in this case because it would require choosing arbitrary thresholds to determine finite groups of genes that are associated with deeper quiescence (i.e negatively correlated with pseudotime) and/or expressed at meaningful levels in mouse cell clusters. We believe that it is not possible to rigorously define these thresholds and we have thus preferred to use non-quantitative terms in the text.

For information, however: we have generated plots displaying the proportion of genes negatively correlated with pseudotime (for different cutoffs of correlation coefficient) that are expressed in either astrocytes, or RGLs or both in the dataset from Cebrian-Silla et al for different cutoffs of what constitutes meaningful expression.

Regarding the cutoffs that we believe are the most justifiable based on how they select known markers (a correlation coefficient below -0.1 to select genes negatively correlated with pseudotime and a cutoff of average normalized expression of 0.5 to accept that a gene is expressed in a cluster), the proportion of genes expressed in both qRGL and Astrocytes is 51%, the proportion of genes expressed in Astrocytes is 56% and the proportion of genes expressed in qRGL is 56%. This translates to 317 genes out of 567 being expressed in either astrocytes or qRGL and 289 genes being expressed in both (and thus 56 that are uniquely expressed in either cluster without being expressed in the other). Adding a cutoff for expression in our zebrafish dataset would also further increase percentages of genes that are found to be expressed in mouse, but at the cost of adding another arbitrary threshold.

- line 168: mention how the mapping of zebrafish glia to mammalian data was done.

We have now added a mention of the MetaNeighbor algorithm which was used for mapping (text l.183).

3.2. some statements in abstract and main text seem exaggerated and unsupported by the data; examples

-- line 60: "full extent of cell diversity"; the dataset is small and does not capture the full diversity of glutamatergic and GABAergic neurons. I could not find anywhere a statement on how many cells were sequenced from zebrafish.

This is now indicated (Text l.52-54).

-- "comprehensive" in line 269 also seems an overstatement (if it refers to neurons)

We apologize for these formulations that may have come across as us trying to overstate our results and we have modified these sentences as suggested.

3.3. Definitions

Providing exact definitions for the terms used would be useful. Examples: what is a "pre-activated RG"? (line 120); what definition of "astrocyte" are the authors following? (line 164 states that besides mammals most species are thought not to have astrocytes, but other scientists had referred to zebrafish glial cells as astrocytes...)

- Pre-activated RG: We have included an explanation of how the different cells were grouped under these terms for Figure 3 (main text l.128-130, new Figure 3 and its legend), which we hope helps clarify our nomenclature.

- Astrocyte: We mean here a cell that has features of an astroglia and has lost its stem cell potential under physiological conditions. Throughout the paper we mostly refer to parenchymal astrocytes, but we do consider the possibility of ventricular astrocytes (in particular in the context of the clonal

analysis) and we have attempted to specify it whenever necessary. Under this definition, zebrafish are not thought to have any astrocytes (and our results are in line with this belief). Less restrictive definitions have indeed been used in other publications (for example Jurisch-Yaksi et al., *Glia* 2020, or Lust et al., *Science* 2022), which include neural stem cells and even in some cases ependymocytes. We have now added the precision that we consider that RG are the only cells to act as neural stem cells in the introduction (l.37-38) to clarify the distinction that we make between RG and astrocytes.

3.4. Content and clarity of the main figures

- figure 1b: Add axes to the figure to indicate this is a tSNE

This has been done.

- the labels of the ancestors in figure 3 are confusing

In our previous version of Figure 3 we were trying to reconstruct the state of the neurogenic cascade in common ancestors of the different species we surveyed, partly to mitigate the risk of drop out due to low sensitivity of scRNAseq. For example, *Ascl1* is not detected in our *Ambystoma mexicanus* dataset and detected at low levels in our *Pleurodeles waltl* dataset. We are unsure whether this means that *Ascl1* really isn't expressed at any stage in the neurogenic cascade in salamanders or whether this is due to a technical issue. However, since we do detect *Ascl1* expression in paRG in zebrafish, *Pogona vitticeps* and mouse, we could reasonably assume that it was expressed in the same way in the tetrapod ancestor.

However, to follow our Referee's advice, in our new version of Figure 3 we now directly summarize the expression of genes of interest without attempting to reconstruct the state of their expression in ancestral species, leaving it to the reader for interpretation. We have thus removed the ancestor labels.

- indicate cluster number in figure S4

This has been done.

- figure 5: at the very least, add a figure legend to indicate what the colors mean (the information is in the caption, but having a legend in the figure itself makes it easier to read).

This has been done.

In general, I thought that the figures include too many summary diagrams and that all the interesting data is in the supplementary file. The authors may want to consider moving some of the supplementary figures, in particular figures 18 and 19, to the main figures.

We felt that our initial Fig.S18 and S19 were themselves schemes. Therefore we chose to modify Fig.4 instead, by adding to it the lineage tracing data.

Referee 2

In the manuscript entitled "Integrative single-cell transcriptomics clarifies adult neurogenesis and macroglia evolution", Morizet et al. used zebrafish model and a very comprehensive metanalysis of transcriptomic data across a wide range of animal species to address the evolutionary changes in neuronal and glial cell types, in search for homologies and differences.

Overall, the authors conducted a thorough and meticulous computational analysis of their single cell transcriptomic dataset, as well as they reanalyzed previously published datasets. Multiple control analyses were performed to ensure the accuracy of clustering. Utilizing their computational workflow, the authors were able to uncover novel insights from published data and examine the topic with an

unparalleled level of resolution. The text is solid, concisely and logically written, technically sound, and most conclusions/interpretations justified by the results obtained.

The results of this work will surely increase our current knowledge of neural cell type (both glial and neuronal) evolutionary aspects and heterogeneity.

In my view, it could be published with some minor revisions.

Here, below, I have some advice:

Main:

The Authors propose/show that features of neural progenitors, especially neurogenic processes, can be highly conserved through evolution (Author's text: "ventricular progenitor patterning is maintained not only throughout life but also throughout evolution"). Then, in supplementary material they address the remarkable differences existing among orders/species, as well as among mammals, especially referring to the remarkable decrease of adult neurogenesis in humans (generally large-brained species) with respect to laboratory rodents. I think that both these statements are true, I'm just wondering whether the reader might make confusion between the well conserved features of neurogenesis (both in ontogeny and in evolution) and the differences that occur among species. Maybe this should be made clearer in the main text (not only in supplementary), by underlying such difference. I say this because, in the current controversy whether human hippocampal neurogenesis persists or not in the adult, many people confuse (or use) the evolutionary conservation of the processes/progenitors/genes to support the general notion that existence/persistence of adult neurogenesis in humans can be true or logically acceptable. In other words: the fact that neurogenic processes are conserved through evolution does not mean that their occurrence, location, and rate must be the same in different species.

We now specify in the main text that, when considering gene expression in ventricular cells, we are exclusively comparing zebrafish and mouse (l. 109) (see also Discussion l. 312-313). In addition, we now distinguish the molecular conservation of the neurogenesis cascade with that of neurogenic sites (Discussion l.326-328).

Minor points:

- SEZ is an old terminology, usually replaced by SVZ

We replaced all mentions of SEZ by SVZ.

Supplementary information:

- Lines 20 and following (paragraph: Homology of neurogenic niches)

For general readers maybe it's better to remind/explain the homologies/differences occurring in the regions of the olfactory system and hippocampus between zebrafish and mammals

We have now added a mention of the architectural similarities in the olfactory systems of zebrafish and mammals, as well as some of the functional differences (Supp. Text l. 44-54).

We also refer to the description of a region involved in spatial orientation in zebrafish in an area that is expected to be an ontogenetic homolog of the hippocampus according to the eversion model for zebrafish brain development (Supp. Text l. 66).

- Line 73-76: Author's text: "...which will shed light on the origin of cell type diversification in the hippocampus. This is of particular interest for the study of adult neurogenesis as the dentate gyrus niche has been described as being more of a mammalian innovation²³ " This thesis by Kempermann was considered wrong by evolutionists (see Commentary by Powers A Brain Behav Evol. 2013, on another review article by Kempermann, but on a similar concept).

Indeed the thesis developed by G. Kempermann has been criticized in the past and we do not subscribe to it either. We have made modifications to make both of these elements clearer in the text (Supp. Text I. 83-86), in particular by specifically mentioning similarities between RG patterning and response to pro-neurogenesis stimuli across species (arguing against the idea of a brutal shift in the mammalian lineage) and that both the SGZ and SVZ have modified architecture and cellular environment (arguing against the idea that the SGZ specifically underwent modifications while the SVZ niche is vestigial).

- Lines 89-91: Author's text: "... Thus, although there is some evidence that immature neurons can still be detected in adult humans, adult neural stem cells remain elusive, and it is more largely agreed that they are likely prematurely depleted compared to most other species 37"

Maybe the concept of "immature" neurons as I think it is intended here (non-newlyborn, "dormant" neurons, such as those described in the cortex layer II) should be explained to the reader. The current, provisional word "immature neuron" can be confused with the newlyborn neurons produced from active stem cell niches in adult neurogenesis; they also share a phase of immaturity with expression of the same markers.

We indeed wanted to refer to the existence of dormant immature neurons as a potential confounder for studies that have concluded that adult neurogenesis persists based only on immature neuron markers. Our original text lacked clarity and we have modified it to mention those dormant neurons in a more direct way (Supp. Text I. 99-103).

- Lines 109-115: Author's text: "...Contrary to arguments explaining the low neurogenesis in humans based on a selection against plasticity due to the size and complexity of their brain⁴⁶, this interpretation suggests a tradeoff with an expanded neocortex." I do not think that this is "contrary". The brain size and neocortex expansion are usually correlated. And this does not exclude the tradeoff with expanded neocortex.

We believe that the explanation we propose differs from the one offered by Pasko Rakic and colleagues when it comes to determining what might have driven premature termination of neurogenesis. In our model, we do not assume that neurogenesis –and to a larger extent plasticity– is detrimental and must have been selected against to improve cognitive functions in the course of human evolution. Instead, we propose that the developmental molecular processes involved in maintaining neural stem cells are to some extent at odds with those that allowed expansion of the neocortex in humans and that the benefits of an expanded neocortex outweighed the putative benefits of extended adult neurogenesis. In this case, instead of interpreting lowered adult neurogenesis as a direct method of optimization of cognitive function, we view it first as an indirect consequence of neocortex expansion. Thus: (i) the model of Pasko Rakic and colleagues makes an assumption on the effect of adult neurogenesis on cognition (namely, that neurogenesis is detrimental and that plasticity must have been selected against to promote intelligence and long term memories) without making a direct assumption about a developmental relationship between premature cessation of adult neurogenesis and expansion of the neocortex (the two could have happened independently, with both being beneficial); in contrast, (ii) our model makes an assumption on the relationship between maintenance of adult neurogenesis and expansion of the neocortex (because the same molecular mechanism relying on the Notch pathway has opposite effects on both, increasing one leads to a decrease in the other) but not on the absolute value of adult neurogenesis (it can be either beneficial or detrimental, but an expanded neocortex in the living conditions of early humans was more beneficial and thus selected for).

Previous studies, cited in the text, support our hypothesis about the effects of the Notch pathway and the expected consequences of a change from NOTCH3 to NOTCH2 on its level of activity. On the other hand, we believe that a few lines of evidence further argue against the thesis developed by Pasko Rakic and colleagues :

- 1) The discovery of immature but non-newly-born neurons that are enriched in large-brained mammals suggests that maintaining plasticity into adulthood does not necessarily go against the development of an increased brain size and/or function;
- 2) In silico modeling of the impact of neurogenesis on network function (see for example https://tspace.library.utoronto.ca/bitstream/1807/109139/4/Tran_Lina_202111_PhD_thesis.pdf) suggests that ongoing neurogenesis promotes both de novo learning and forgetting, which is consistent with experimental results and does not support the idea that ongoing neurogenesis is detrimental for network function;
- 3) One of the main arguments in favor of adult neurogenesis being detrimental is that this added plasticity prevents formation of long-term memories. However in addition to the in silico models mentioned above, work on memory encoding and consolidation suggests that a hippocampo-cortical coupling through specific coordinated neuronal activity rhythms allows for memory consolidation in the cortex (see for example <https://www.nature.com/articles/nn.4304>). Thus, old memories are in large part encoded in the cortex and only precise contextual memories seem to also rely on hippocampal activity after a while (see for example <https://www.ncbi.nlm.nih.gov/pmc/articles/PMC2928141/>).

Therefore, although the two models can yield similar predictions, they rely on different assumptions. We believe that our model is simpler, less human-centered and has more support in the literature. The two models also have opposite implications on the potential interest of trying to reactivate adult neurogenesis, which we believe is an important distinction.

We modified our sentence so as not to take such strong position as in our initial text (both models being speculative at this point) but have also tried to better specify that the difference lies in the fact that our model is based on a developmental process rather than repression of adult neurogenesis function (Supp. Text I. 123-126).

Author's text: "Selection for these traits would have taken place at a moment when sight was already the primary sensory modality thus reducing selective pressure on olfactory processing⁴⁷, and when life expectancy was much shorter, thus resulting in only a short period of time without hippocampal neurogenesis given that it persists well into teenage years." This is also supported by work of Aboitiz & Montiel on the origin of isocortex.

We must admit that we were not aware of this work, and we apologize for this. The thesis developed regarding the relationship between the size of the limbic system and of the isocortex is indeed very interesting and relevant and we have now included a sentence to discuss it and cite the 2005 paper from Aboitiz and Montiel entitled "Olfaction, navigation, and the origin of isocortex" (ref.54). We thank the reviewer for pointing this out and giving us the opportunity to read the work of these authors and to correct our oversight by adding this point (Supp. Text I. 130-135).

- Line 170 and following: Author's text: "we scored mouse astroglial cells for genes enriched in q4 over q2"

The authors compare the cells in the q4 and q2 clusters to identify markers that resemble the transcriptional signature of astrocytes. However, it is not entirely clear why they selected these two clusters over the other dorsal qRG clusters (q1, q3). The authors should provide a better explanation and justification for their selection.

We have now expanded our explanation as to why we focused on q4 and q2 for this comparison (Main text I. 186-190). The ISH for *draxin*, which labels q3 among RGs, was somewhat regionalized and we wanted to mitigate the risk of recovering a region-specific signature. We used q2 instead of q1 because q1 already has a transcriptomic signature of cells close to proliferation and/or commitment, and we feared that this might artificially enhance the differences we expected to see, as cells turn off glial

markers when they activate. Nonetheless the results remain very similar if instead we look for differentially expressed genes between q4 and q1 or between the two clusters that map more strongly to astrocytes (q3 and q4) and the two clusters that map more strongly to RG (q1 and q2) (see Fig.R2.1 below).

Figure R2.1. Number (and list) of differentially expressed genes between the q clusters indicated.

- The authors frequently refer to the expression levels of specific genes across and within different species in the manuscript. However, they often do not present absolute expression levels in the form of plots or tables. Instead, they provide representations of relative expression levels or schematic summaries (e.g., Figure 3, Figure 5, etc.). To support their discussion, the authors should include through the whole manuscript evidence of the expression levels of the genes they refer to in the form of tables with raw data or expression plots.

We have now added several files to our online repository displaying the expression of several genes that were used to draw our conclusions, in particular for Figs.3 and 5 (see <https://entrepot.recherche.data.gouv.fr/privateurl.xhtml?token=bede1d62-f7cf-4e85-b6ce-05121176f108>).

We have also generated datasets and made them available in a list format that is part of the default R installation and is thus compatible with any R version. This format requires minimal coding, and we also provide functions that can be used to replicate the expression plots or plot other genes of interest by simply copy-pasting them and replacing the gene names.

We include in parallel in the manuscript a more complete description of what this online repository contains (Main text l.633-645): “This site includes rds files for our dataset as well as most of the re-analyzed datasets unless. Re-analyzed datasets were not included only if 1) they are already readily available through the original publication, 2) our analyses agree with that of the original publication and 3) we did not conduct analyses requiring further partition of the original data. Although most of the analyses were conducted using Seurat v2.3.4 to store the data, we provide them in list format with accompanying custom plotting functions so that they can be downloaded and used with minimal coding experience and independently of the specific local R installation. We also provide html file to directly query the expression of genes relevant in our analyses, in particular the reconstruction of gene expression along the neurogenic cascade and the identification of cells expressing genes belonging to

the astrocytic geneset, across the different datasets. Finally, the site includes a README file detailing the different datasets there, how to download them and how to extract raw or normalized data and/or plot genes of interest”.

Finally, we extensively modified Fig.3 so as to directly show expression plots.

- Line 179 and following: The authors should provide a more comprehensive explanation for the clonal analysis and clarify in the main text that the labeling of Her4 cells will not be exclusive to q4 cells but will also include cells from other clusters. This is apparent from the legend of Extended Data Figure 14, but it is not adequately explained in the main text. Additionally, it is unclear from the text when the proportion of q4 cells is estimated. Based on the text, it appears to be at 507 days of chase, but it should instead coincide with the time of Tx administration. Moreover, how can the authors conclude that the difference is not due to cell death?

We initially did not have images combining clonal labeling with RNAScope smFISH staining, and our reasoning relied on convergent arguments:

- 1) *her4.3* (driving ERT2CreERT2) is expressed in virtually all qNSCs, and at a higher level in q4 compared to q2;
- 2) The estimated proportion of q4 cells, obtained independently from scRNAseq and rough *in situ* quantification based on smFISH for *timp4.3* is around 50% (this is at 3mpf, hence at the onset of the experiment, we now specify this in the text -legend to Fig.4-). Although both methods for estimating the proportion of q4 cells are imperfect, they give similar results;
- 3) The proportion of clones that do not give rise to neurons and might thus not have neurogenic potential is at most slightly less than 5%;
- 4) Finally, we previously demonstrated that there is virtually no cell death in the zebrafish pallium (see Than trong et al, Sci Adv, 2020).

However, our Referee is right in that there might have been a bias in induction due to factors independent of *her4.3* expression that would lead to selection against q4 cells upon clonal induction and invalidate this reasoning. Thus, to answer this question we finally managed to obtain joint staining for clonal induction and *timp4.3* smFISH, at the earliest timepoint where reporter signal can be seen (6 days post-induction), as requested. This experiment showed that *timp4.3* expression is comparable in labeled cells and in control (non-labeled) cells (new Fig.4e-f, new Fig.S15a-d).

A further control to this quantification is provided by scoring *timp4.3* expression based on the number of cells per clone: although most labeled cells remain quiescent for long periods, a few do divide during the 6 days between induction and “time point zero” analysis at 6dpi, generating 2- or 4-cell clones. We found that *timp4.3* expression levels decrease with increasing divisions, as expected from RG progression along the lineage (Fig.S15c).

Together, these results show that q4 cells are selected by the induction in proportion similar to their proportion in the total RG population. Therefore, this analysis of *timp4.3* expression in clonal cells at 6dpi reinforces our initial conclusion, and we thank the reviewer for providing the necessary incentive to carry out this experiment.

- Line 236: please, provide original data in table or plot for this statement.

Now l.241-252. We have now added several files to our data.gouv repository (<https://entrepot.recherche.data.gouv.fr/privateurl.xhtml?token=bede1d62-f7cf-4e85-b6ce-05121176f108>) containing plots for detected homologs of the genes of the astrocytic dataset (Table S1) in representative datasets, as well as a description in the README file of how to easily extract the data if desired.

- Extended data figure 3: it is not clear here how the two plots should be compared. The plots here should be more informative to make the comparison clearer also for non-experts.

We have added explanations in the figure legend.

- Extended data figure 6: Clusters classification should be indicated in the plots to simplify the understanding of the plot, similarly to figure 17

We believe our Referee was referring to Figure S4. We have added cluster numbers on the plots.

- One notable advantage of this study compared to others is the final number of cells included in the analysis after all filtering steps. Therefore, it would be beneficial to clearly state this number in the main text

We now do so (17,710 cells were analyzed in our dataset, and over 2,000,000 cells from other datasets in different species (Main text I.52-54).

REVIEWERS' COMMENTS

Reviewer #1 (Remarks to the Author):

In this revised version of the manuscript, the authors addressed thoroughly all the questions and concerns of the reviewers.

The revised figure 3 is a nice addition that gives a concrete idea of the primary data on which the conclusions are based on.

Minor comments I have are:

1. in figure 3, the figure legend showing the color scheme for counts is too small and impossible to read. Please adjust.
2. During the revision of this manuscript, at least two other papers on the fish telencephalon have been published in peer-reviewed journals, plus a couple others were posted as preprints in biorxiv (there might be even more that escaped my attention). The published papers are Tibi et al Science Advances and Anneser et al Current Biology. Those papers describe neuron types in the fish telencephalon in depth, and should be cited by the authors when they report on GABAergic neurons in zebrafish.

Reviewer #1 (Remarks on code availability):

A README file is provided

All code is reported in R markdown format, which is reproducible and seems correct (although I haven't tried to run it myself). I should say that the analyses reported in this paper are relatively simple from a computational point of view.

All the datasets necessary for reproducing the analyses are provided

Instructions are clear and this code repository is a usable resource for the community

Reviewer #2 (Remarks to the Author):

The Authors have answered to the my comments/modified the text accordingly in a complete way. They have also answered (in my opinion very well) to the comments of the other Reviewer. Hence, I think that now the manuscript is suitable for publication in Nat Commun

Response to reviewers:

Reviewer #1 (Remarks to the Author):

In this revised version of the manuscript, the authors addressed thoroughly all the questions and concerns of the reviewers. The revised figure 3 is a nice addition that gives a concrete idea of the primary data on which the conclusions are based on.

Minor comments I have are:

1. in figure 3, the figure legend showing the color scheme for counts is too small and impossible to read. Please adjust.

→ The font has been enlarged

2. During the revision of this manuscript, at least two other papers on the fish telencephalon have been published in peer-reviewed journals, plus a couple others were posted as preprints in biorxiv (there might be even more that escaped my attention). The published papers are Tibi et al Science Advances and Anneser et al Current Biology. Those papers describe neuron types in the fish telencephalon in depth, and should be cited by the authors when they report on GABAergic neurons in zebrafish.

→ We have added these references (text l.85-88 and l.328-331, ref. nb 25 and 26).

Reviewer #1 (Remarks on code availability):

A README file is provided

All code is reported in R markdown format, which is reproducible and seems correct (although I haven't tried to run it myself). I should say that the analyses reported in this paper are relatively simple from a computational point of view.

All the datasets necessary for reproducing the analyses are provided

Instructions are clear and this code repository is a usable resource for the community

Reviewer #2 (Remarks to the Author):

The Authors have answered to the my comments/modified the text accordingly in a complete way. They have also answered (in my opinion very well) to the comments of the other Reviewer.

Hence, I think that now the manuscript is suitable for publication in Nat Commun